# Bridging Geometric States
# via Geometric Diffusion Bridge

**Shengjie Luo**[1,*], **Yixian Xu**[1,4*], **Di He**[1†], **Shuxin Zheng**[2], **Tie-Yan Liu**[2], **Liwei Wang**[1,3†]

[1]State Key Laboratory of General Artificial Intelligence,
School of Intelligence Science and Technology, Peking University
[2]Microsoft Research AI4Science   [3]Center for Data Science, Peking University
[4]Pazhou Laboratory (Huangpu), Guangzhou, Guangdong 510555, China
`luosj@stu.pku.edu.cn`, `xyx050@stu.pku.edu.cn`,
`{shuz, tyliu}@microsoft.com`, `{dihe, wanglw}@pku.edu.cn`

## Abstract

The accurate prediction of geometric state evolution in complex systems is critical for advancing scientific domains such as quantum chemistry and material modeling. Traditional experimental and computational methods face challenges in terms of environmental constraints and computational demands, while current deep learning approaches still fall short in terms of precision and generality. In this work, we introduce the Geometric Diffusion Bridge (GDB), a novel generative modeling framework that accurately bridges initial and target geometric states. GDB leverages a probabilistic approach to evolve geometric state distributions, employing an equivariant diffusion bridge derived by a modified version of Doob's $h$-transform for connecting geometric states. This tailored diffusion process is anchored by initial and target geometric states as fixed endpoints and governed by equivariant transition kernels. Moreover, trajectory data can be seamlessly leveraged in our GDB framework by using a chain of equivariant diffusion bridges, providing a more detailed and accurate characterization of evolution dynamics. Theoretically, we conduct a thorough examination to confirm our framework's ability to preserve joint distributions of geometric states and capability to completely model the underlying dynamics inducing trajectory distributions with negligible error. Experimental evaluations across various real-world scenarios show that GDB surpasses existing state-of-the-art approaches, opening up a new pathway for accurately bridging geometric states and tackling crucial scientific challenges with improved accuracy and applicability.

## 1 Introduction

Predicting the evolution of the geometric state of a system is essential across various scientific domains [46, 88, 55, 17, 20, 101], offering valuable insights into difficult tasks such as drug discovery [25, 29], reaction modeling [9, 24], and catalyst analysis [13, 105]. Despite its critical importance, accurately predicting future geometric states of interest is challenging. Experimental approaches often face obstacles due to strict environmental requirements and physical limits of instruments [102, 3, 69]. Computational approaches seek to solve the problem by simulating the dynamics based on underlying equations [81, 88]. Though providing greater flexibility, such calculations are typically driven by first-principle methods or empirical laws, either requiring extensive computational costs [68] or sacrificing accuracy [40].

---

[*]Equal contribution.
[†]Correspondence to: Di He<`dihe@pku.edu.cn`>, Liwei Wang <`wanglw@pku.edu.cn`>.

38th Conference on Neural Information Processing Systems (NeurIPS 2024).

In recent years, deep learning has emerged as a pivotal tool in scientific discovery for many fields [43, 23, 69, 107], offering new avenues for tackling this problem. One line of approach aims to train models to predict target geometric states (e.g., equilibrium states) from initial states directly and develop neural network architectures that respect inherent symmetries of geometric states, such as the equivariance of rotation and translation [104, 31, 8, 87, 89, 103]. However, this paradigm requires encoding the iterative evolution into a single-step prediction model, which lacks the ability to fully capture the system's underlying dynamics and potentially leading to reduced accuracy. Another line of research trains machine learning force fields (MLFFs) to simulate the trajectory of geometric states over time [32, 34, 6, 70, 5, 58], showing a better efficiency-accuracy balance [15, 13, 105, 84]. Nevertheless, MLFFs are typically trained to predict intermediate labels, such as the force of the (local) current state. During inference, states are iteratively updated step by step. Since small local errors can accumulate, reliable predictions over long trajectories highly depend on the quality of intermediate labels, which cannot be guaranteed [7, 106, 30]. Therefore, an ideal solution that can precisely bridge initial and target geometric states and effectively leverage trajectory data (if available) as guidance is in great demand.

In this work, we introduce **Geometric Diffusion Bridge (GDB)**, a general framework for bridging geometric states through generative modeling. From a probabilistic perspective, predicting target geometric states from initial states requires modeling the joint state distribution across different time steps. The diffusion models [37, 99] are standard choices to achieve this goal. However, these methods ideally generate data by denoising samples drawn from a Gaussian prior distribution, which makes it challenging to bridge pre-given geometric states or leverage trajectories in a unified manner. To address the issue, we establish a novel *equivariant diffusion bridge* by developing a modified version of *Doob's $h$-transform* [82, 81, 16]. The proposed stochastic differential equation (SDE) is anchored by initial and target geometric states to simultaneously model the joint state distribution and is governed by equivariant transition kernels to satisfy symmetry constraints. Intriguingly, we further demonstrate that this framework can seamlessly leverage trajectory data to improve prediction. With available trajectory data, we can construct *chains of equivariant diffusion bridges*, each modeling one segment in the trajectory. The segments are interconnected by properly setting the boundary conditions, allowing complete modeling of trajectory data. For model training, we derive a scalable and simulation-free matching objective similar to [59, 61, 77], which requires no computational overhead when trajectory data is leveraged.

Overall, our GDB framework offers a unified solution that precisely bridges geometric states by modeling the joint state distribution and comprehensively leverages available trajectories as fine-grained depiction of dynamics for enhanced performance. Mathematically, we prove that the joint distribution of geometric states across different time steps can be completely preserved by our (chains of) equivariant diffusion bridge technique, confirming its expressiveness in bridging geometric states and underscoring the necessity of design choices in our framework. Furthermore, under mild and practical assumptions, we prove that our framework can approximate the underlying dynamics governing the evolution of geometric state trajectories with negligible error in convergence, remarking on the completeness and usefulness of our framework in different scenarios. These advantages show the superiority of our framework over existing approaches.

Practically, we provide a comprehensive guidance for implementing our GDB framework in real-world applications. To verify its effectiveness and generality, we conduct extensive experiments covering diverse data modalities (simple molecules & adsorbate-catalyst complex), scales (small, medium and large scales) and scenarios (with & without trajectory guidance). Numerical results show that our GDB framework consistently outperforms existing state-of-the-art machine learning approaches by a large margin. In particular, our method even surpasses strong MLFF baselines that are trained on $10\times$ more data in the challenging structure relaxation task of OC22 [105], and trajectory guidance can further enhance our performance. The significantly superior performance demonstrates the high capacity of our framework to capture the complex evolution dynamics of geometric states and determine valuable and crucial geometric states of interest in critical real-world challenges.

## 2 Background

### 2.1 Problem Definition

Our task of interest is to capture the evolution of geometric states, i.e., predicting future states from initial states. Formally, let $S$ denote a system consisting of a set of objects located in the

three-dimensional Euclidean space. We use $\mathbf{H} \in \mathbb{R}^{n \times d}$ to denote the objects with features, where $n$ is the number of objects, and $d$ is the feature dimension. For object $i$, let $\mathbf{r}_i \in \mathbb{R}^3$ denote its Cartesian coordinate. We define the system as $S = (\mathbf{H}, R)$, where $R = \{\mathbf{r}_1, ..., \mathbf{r}_n\}$. This data structure ubiquitously corresponds to various real-world systems such as molecules and proteins [17, 20, 101]. In practice, the geometric state is governed by physical laws and evolves over time, and we denote the geometric state at a given time $t$ as $R^t = \{\mathbf{r}_1^t, ..., \mathbf{r}_n^t\}$. Given a system $S^{t_0} = (\mathbf{H}, R^{t_0})$ at time $t_0$, our goal is to predict $S^{t_1} = (\mathbf{H}, R^{t_1})$ at a future time $t_1$. As an example, in a molecular system, $R^{t_1}$ can be the equilibrium state of interest evolved from the initial state $R^{t_0}$.

In this problem, inherent symmetries in geometric states should be considered. For example, a rotation that is applied to the coordinate system at time $t_0$ should also be applied to subsequent time steps. These symmetries are related to the concept of equivariance in group theory [19, 18, 91]. Formally, let $\phi : \mathcal{X} \to \mathcal{Y}$ denote a function mapping between two spaces. Given a group $G$, let $\rho^{\mathcal{X}}$ and $\rho^{\mathcal{Y}}$ denote its group representations, which describe how the group elements act on these spaces. A function $\phi : \mathcal{X} \to \mathcal{Y}$ is said to be equivariant if it satisfies the following condition: $\rho^{\mathcal{Y}}(g)[\phi(x)] = \phi\left(\rho^{\mathcal{X}}(g)[x]\right), \forall g \in G, x \in \mathcal{X}$. When $\rho^{\mathcal{Y}} = \mathcal{I}^{\mathcal{Y}}$ (identity transformation), it is also known as invariance. SE(3) group, which pertains to translations (T(3)) and rotations (SO(3)) in 3D Euclidean space, is one of the most widely used groups and is employed in our framework.

## 2.2 Diffusion Models

Diffusion models [95, 37, 99] have emerged as the state-of-the-art generative modeling approaches across various domains [83, 85, 47, 115, 113, 117]. The main idea of this method is to construct a diffusion process that maps data to noise, and train models to reverse such process by using a tractable objective.

Formally, to model the data distribution $q_{data}(\mathbf{X})$, where $\mathbf{X} \in \mathbb{R}^d$, we construct a diffusion process $(\mathbf{X}_t)_{t \in [0,T]}$, which is represented as a sequence of random variables indexed by time steps. We set $\mathbf{X}_0 \sim q_{data}(\mathbf{X})$ and $\mathbf{X}_T \sim p_{prior}(\mathbf{X})$, where $p_{prior}(\mathbf{X})$ has a tractable form to generate samples efficiently, e.g. standard Gaussian distribution. Mathematically, we model $(\mathbf{X}_t)_{t \in [0,T]}$ as the solution to the following stochastic differential equation (SDE):

$$\mathrm{d}\mathbf{X}_t = \mathbf{f}(\mathbf{X}_t, t)\mathrm{d}t + \sigma(t)\mathrm{d}\mathbf{B}_t, \tag{1}$$

where $\mathbf{f}(\cdot, \cdot) : \mathbb{R}^d \times [0, T] \to \mathbb{R}^d$ is a vector-valued function called the *drift* coefficient, $\sigma(\cdot) : [0, T] \to \mathbb{R}$ is a scalar function known as the *diffusion* coefficient, and $(\mathbf{B}_t)_{t \in [0,T]}$ is the standard Wiener process (a.k.a., Brownian motion) [26]. We hereafter denote by $p_t(\mathbf{X})$ the marginal distribution of $\mathbf{X}_t$. Let $p(x', t'|x, t)$ denote the transition density function such that $P(\mathbf{X}_{t'} \in A | \mathbf{X}_t = x) = \int_A p(x', t'|x, t)\mathrm{d}x'$ for any Borel set $A$. By simulating this diffusion process forward in time, the distribution of $\mathbf{X}_t$ will become $p_{prior}(\mathbf{X})$ at the final time $T$. In the literature, there exist various design choices of the SDE formulation in Eqn. (1) such that it transports the data distribution into the fixed prior distribution [98, 37, 99, 72, 97, 47].

In order to sample $\mathbf{X}_0 \sim p_0(\mathbf{X}) := q_{data}(\mathbf{X})$, an intriguing fact can be leveraged: the reverse of a diffusion process is also a diffusion process [2]. This reverse process runs backward in time and can be formulated by the following time-reversal SDE:

$$\mathrm{d}\mathbf{X}_t = \left[\mathbf{f}(\mathbf{X}_t, t) - \sigma^2(t)\nabla_{\mathbf{X}_t} \log p_t(\mathbf{X}_t)\right] \mathrm{d}t + \sigma(t)\mathrm{d}\mathbf{B}_t, \tag{2}$$

where $\nabla_{\mathbf{X}} \log p_t(\mathbf{X})$ denote the score of the marginal distribution at time $t$. If the score is known for all time, then we can derive the reverse diffusion process from Eqn. (2), sample from $p_{prior}(\mathbf{X})$, and simulate this process to generate samples from the data distribution $q_{data}(\mathbf{X})$. In particular, the score $\nabla_{\mathbf{X}} \log p_t(\mathbf{X})$ can be estimated by training a parameterized model $\mathbf{s}_\theta(\mathbf{X}, t)$ with a denoising score matching objective [98, 97]. In theory, the minimizer of this objective approximates the ground-truth score [99] and this objective is tractable.

## 3 Geometric Diffusion Bridge

As discussed in the introduction, effectively capturing the evolution of geometric states is crucial, for which three desiderata should be carefully considered:

- *Coupling Preservation*: From a probabilistic perspective, the evolution of geometric states transports their distribution from $q_{\text{data}}(S^{t_0})$ to $q_{\text{data}}(S^{t_1})$, and we are interested in modeling the distribution of target geometric states given the initial states, i.e., $q_{\text{data}}(S^{t_1}|S^{t_0}) := q_{\text{data}}(R^{t_1}|\mathbf{H}, R^{t_0})$, which can be achieved by preserving the *coupling* of geometric states, i.e., $q_{\text{data}}(R^{t_0}, R^{t_1}|\mathbf{H})$. For brevity, we hereafter omit the condition of $\mathbf{H}$ because it keeps the same along the evolution and can be easily incorporated into the models.

- *Symmetry Constraints*: Since the law governing the evolution is unchanged regardless of how the system is rotated or translated, the distribution of the geometric states should satisfy symmetry constraints, i.e., $q_{\text{data}}(\rho^{\mathcal{R}}(g)[R^{t_1}]|\rho^{\mathcal{R}}(g)[R^{t_0}]) = q_{\text{data}}(R^{t_1}|R^{t_0})$ and $q_{\text{data}}(\rho^{\mathcal{R}}(g)[R^{t_0}], \rho^{\mathcal{R}}(g)[R^{t_1}]) = q_{\text{data}}(R^{t_0}, R^{t_1})$ for all $g \in \text{SE}(3), R^t \in \mathcal{R}$.

- *Trajectory Guidance*: Trajectories of geometric states are sometimes accessible and provide fine-grained descriptions of the evolution dynamics. For completeness, it is crucial to develop a unified framework that can characterize and leverage trajectory data as guidance for better bridging geometric states and capturing the evolution.

However, existing approaches typically have their limitations for this task, which we thoroughly discuss in Sec. 5 and summarize into Table 1. In this section, we introduce Geometric Diffusion Bridge (GDB), a general framework for bridging geometric states through generative modeling. We will elaborate on key techniques for completely preserving couping under symmetry constraints (Sec. 3.1), and demonstrate how our framework can be seamlessly extended to leverage trajectory data (Sec. 3.2). Theoretically, we conduct a thorough analysis on the capability of our unified framework, showing its completeness and superiority. All proofs of theorems are presented in Appendix B. A detailed guidance of practical implementing our framework is further provided (Sec. 3.3).

Table 1: Comparisons of different candidates for bridging geometric states

| Methods | Symmetry Constraints | Coupling Preservation | Trajectory guidance |
|---|:---:|:---:|:---:|
| Direct Prediction [104, 31, 87, 89, 8] | ✓ | ✓ | ✗ |
| MLFFs [90, 33, 6, 34, 58] | ✓ | ✗ | ✓ |
| Geometric Diffusion Model [115, 38, 114] | ✓ | ✗ | ✗ |
| **Geometric Diffusion Bridge (ours)** | ✓ | ✓ | ✓ |

## 3.1 Equivariant Diffusion Bridge

Our key design lies in the construction of *equivariant diffusion bridge*, a tailored diffusion process $(\mathbf{R}^t)_{t \in [0,T]}$ for bridging initial states $\mathbf{R}^0 \sim q_{\text{data}}(R^{t_0})$ and target states $\mathbf{R}^T \sim q_{\text{data}}(R^{t_1}|R^{t_0})$, completely preserving coupling of geometric states and satisfying symmetry constraints. Firstly, we investigate necessary conditions for a diffusion process on geometric states to meet the symmetric constraints:

**Proposition 3.1.** *Let $\mathcal{R}$ denote the space of geometric states and $\mathbf{f}_{\mathcal{R}}(\cdot, \cdot) : \mathcal{R} \times [0,T] \to \mathcal{R}$ denote the drift coefficient on $\mathcal{R}$. Let $(\mathbf{W}^t)_{t \in [0,T]}$ denote the Wiener process on $\mathcal{R}$. Given an SDE on geometric states $d\mathbf{R}^t = \mathbf{f}_{\mathcal{R}}(\mathbf{R}^t, t)dt + \sigma(t)d\mathbf{W}^t, \mathbf{R}^0 \sim q(\mathbf{R}^0)$, its transition density $p_{\mathcal{R}}(z', t'|z, t), z, z' \in \mathcal{R}$ is SE(3)-equivariant, i.e., $p_{\mathcal{R}}(\mathbf{R}^{t'}, t'|\mathbf{R}^t, t) = p_{\mathcal{R}}(\rho^{\mathcal{R}}(g)[\mathbf{R}^{t'}], t'|\rho^{\mathcal{R}}(g)[\mathbf{R}^t], t), \forall g \in \text{SE}(3), 0 \leq t, t' \leq T$, if these conditions are satisfied: (1) $q(\mathbf{R}^0)$ is SE(3)-invariant; (2) $\mathbf{f}_{\mathcal{R}}(\cdot, t)$ is SO(3)-equivariant and T(3)-invariant; (3) the transition density of $(\mathbf{W}^t)_{t \in [0,T]}$ is SE(3)-equivariant.*

Using Proposition 3.1, we can obtain a diffusion process that respect symmetry constraints by properly considering conditions for key components. Next, we modify a useful tool in probability theory called *Doob's h-transform* [82, 81, 16], which plays an essential role in the construction of our equivariant diffusion bridge for preserving coupling of geometric states:

**Proposition 3.2.** *Let $p_{\mathcal{R}}(z', t'|z, t)$ be the transition density of the SDE in Proposition 3.1. Let $h_{\mathcal{R}}(\cdot, \cdot) : \mathcal{R} \times [0,T] \to \mathbb{R}_{>0}$ be a smooth function satisfying: (1) $h_{\mathcal{R}}(\cdot, t)$ is SE(3)-invariant; (2) $h_{\mathcal{R}}(z, t) = \int p_{\mathcal{R}}(z', t'|z, t)h_{\mathcal{R}}(z', t')dz'$. Then we can derive the following $h_{\mathcal{R}}$-transformed SDE on geometric states:*

$$d\mathbf{R}^t = \left[\mathbf{f}_{\mathcal{R}}(\mathbf{R}^t, t) + \sigma^2(t)\nabla_{\mathbf{R}^t} \log h_{\mathcal{R}}(\mathbf{R}^t, t)\right] dt + \sigma(t)d\mathbf{W}^t, \qquad (3)$$

*with SE(3)-equivariant transition density $p_{\mathcal{R}}^h(z', t'|z, t)$ equals to $p_{\mathcal{R}}(z', t'|z, t)\frac{h_{\mathcal{R}}(z', t')}{h_{\mathcal{R}}(z, t)}$.*

Proposition 3.2 provides an equivariant version of Doob's $h$-transform, which can be used to guide a free SDE on geometric states to hit an event almost surely. For example, if we set $h_{\mathcal{R}}(\cdot, t) = p_{\mathcal{R}}(z, T | \cdot, t), z \in \mathcal{R}$, i.e., the transition density of the original SDE evaluated at $\mathbf{R}^T = z$, then the $h_{\mathcal{R}}$-transformed SDE in Eqn. (3) arrives at the specific geometric state $z$ almost surely at the final time (see Proposition B.7 in the appendix for more details). Therefore, if we derive a proper $h_{\mathcal{R}}(\cdot, \cdot)$ function under the symmetry constraints, our target process $(\mathbf{R}^t)_{t \in [0, T]}$ can be constructed:

**Theorem 3.3** (Equivariant Diffusion Bridge). *Let* $\mathrm{d}\mathbf{R}^t = \mathbf{f}_{\mathcal{R}}(\mathbf{R}^t, t)\mathrm{d}t + \sigma(t)\mathrm{d}\mathbf{W}^t$ *be an SDE on geometric states with transition density* $p_{\mathcal{R}}(z', t' | z, t), z, z' \in \mathcal{R}$ *satisfying the conditions in Proposition 3.1. Let* $h_{\mathcal{R}}(z, t; z_0) = \int p_{\mathcal{R}}(z', T | z, t) \frac{q_{data}(z' | z_0)}{p_{\mathcal{R}}(z', T | z_0, 0)}\mathrm{d}z'$. *By using Proposition 3.2, we can derive the following* $h_{\mathcal{R}}$-*transformed SDE:*

$$\mathrm{d}\mathbf{R}^t = \left[ \mathbf{f}_{\mathcal{R}}(\mathbf{R}^t, t) + \sigma^2(t)\mathbb{E}_{q_{\mathcal{R}}(\mathbf{R}^T, T | \mathbf{R}^t, t; \mathbf{R}^0, 0)}[\nabla_{\mathbf{R}^t} \log p_{\mathcal{R}}(\mathbf{R}^T, T | \mathbf{R}^t, t) | \mathbf{R}^0, \mathbf{R}^t] \right] \mathrm{d}t + \sigma(t)\mathrm{d}\mathbf{W}^t, \tag{4}$$

*which corresponds to a process* $(\mathbf{R}^t)_{t \in [0, T]}, \mathbf{R}^0 \sim q_{data}(R^{t_0})$ *satisfying the following properties:*

- *let* $q(\cdot, \cdot) : \mathcal{R} \times \mathcal{R} \to \mathbb{R}_{\geq 0}$ *denote the joint distribution induced by* $(\mathbf{R}^t)_{t \in [0, T]}$, *then* $q(\mathbf{R}^0, \mathbf{R}^T)$ *equals to* $q_{data}(R^{t_0}, R^{t_1})$;

- *its transition density* $q_{\mathcal{R}}(\mathbf{R}^{t'}, t' | \mathbf{R}^t, t; \mathbf{R}^0, 0) = q_{\mathcal{R}}(\rho^{\mathcal{R}}(g)[\mathbf{R}^{t'}], t' | \rho^{\mathcal{R}}(g)[\mathbf{R}^t], t; \rho^{\mathcal{R}}(g)[\mathbf{R}^0], 0)$, $\forall 0 \leq t, t' \leq T, g \in \mathrm{SE}(3), \mathbf{R}^0 \sim q_{data}(R^{t_0})$.

*We call the tailored diffusion process* $(\mathbf{R}^t)_{t \in [0, T]}$ *an equivariant diffusion bridge.*

According to Theorem 3.3, given an initial geometric state $R^{t_0}$, we can predict target geometric states $R^{t_1}$ by simulating the equivariant diffusion bridge $(\mathbf{R}^t)_{t \in [0, T]}$ from $\mathbf{R}^0 = R^{t_0}$, which arrives at $\mathbf{R}^T \sim q_{\mathrm{data}}(R^{t_1} | R^{t_0})$. However, the score $\mathbb{E}_{q_{\mathcal{R}}(\mathbf{R}^T, T | \mathbf{R}^t, t; \mathbf{R}^0, 0)}[\nabla_{\mathbf{R}^t} \log p_{\mathcal{R}}(\mathbf{R}^T, T | \mathbf{R}^t, t) | \mathbf{R}^0, \mathbf{R}^t]$ in Eqn. (4) is not tractable in general. Inspired by the score matching objective in diffusion models [99], we use a parameterized model $\mathbf{v}_\theta(\mathbf{R}^t, t; \mathbf{R}^0)$ to estimate the score by using the following training objective:

$$\mathcal{L}(\theta) = \mathbb{E}_{(z_0, z_1) \sim q_{\mathrm{data}}(R^{t_0}, R^{t_1}), \mathbf{R}^t \sim q_{\mathcal{R}}(\mathbf{R}^t, t | z_1, T; z_0, 0)} \lambda(t) \| \mathbf{v}_\theta(\mathbf{R}^t, t; z_0) - \nabla_{\mathbf{R}^t} \log p_{\mathcal{R}}(z_1, T | \mathbf{R}^t, t) \|^2, \tag{5}$$

where $t \sim \mathcal{U}(0, T)$ (the uniform distribution on $[0, T]$), and $\lambda(\cdot) : [0, T] \to \mathbb{R}_{\geq 0}$ is a positive weighting function. Theoretically, we prove that the minimizer of Eqn. (5) approximates the ground-truth score (see Appendix B.5 for more details). Moreover, this objective is tractable because the transition density $p_{\mathcal{R}}$ and $q_{\mathcal{R}}$ can be designed to have simple and explicit forms such as Gaussian, which we will elaborate on in Sec. 3.3.

## 3.2 Chain of Equivariant Diffusion Bridges for Leveraging Trajectory Guidance

In this subsection, we elaborate on how to leverage trajectories of geometric states as a fine-grained guidance in our framework. Let $(\tilde{R}^i)_{i \in [N]}$ denote a trajectory of $N + 1$ geometric states and $q_{\mathrm{traj}}(\tilde{R}^0, ..., \tilde{R}^N)$ denote the joint probability density function of geometric states in a trajectory. In practice, the markov property of trajectories typically holds [109, 78]. Under this assumption, $q_{\mathrm{traj}}(\tilde{R}^0, ..., \tilde{R}^N)$ can be equivalently reformulated into $q_{\mathrm{traj}}^0(\tilde{R}^0) \prod_{i=1}^N q_{\mathrm{traj}}^i(\tilde{R}^i | \tilde{R}^{i-1})$ by the chain rule of probability. If $q_{\mathrm{traj}}^i(\tilde{R}^i | \tilde{R}^{i-1})$ can be well modeled, we can capture the distribution of trajectories of geometric states completely.

According to Theorem 3.3, given $\mathbf{R}^0 \sim q_{\mathrm{traj}}^0(\tilde{R}^0)$, an equivariant diffusion bridge $(\mathbf{R}^t)_{t \in [0, T]}$ can be constructed to model the joint distribution $q_{\mathrm{traj}}(\tilde{R}^0, \tilde{R}^1)$ and hence $q_{\mathrm{traj}}^1(\tilde{R}^1 | \tilde{R}^0)$ is preserved. Therefore, if we construct a series of interconnected equivariant diffusion bridges, the distribution of trajectories can be modeled:

**Theorem 3.4** (Chain of Equivariant Diffusion Bridges). *Let* $\{(\mathbf{R}_i^t)_{t \in [0, T]}\}_{i \in [N-1]}$ *denote a series of* $N$ *equivaraint diffusion bridges defined in Theorem 3.3. For the* $i$-*th bridge* $(\mathbf{R}_i^t)_{t \in [0, T]}$, *if we set (1)* $h_{\mathcal{R}}^i(z, t; z_0) = \int p_{\mathcal{R}}(z', T | z, t) \frac{q_{traj}^{i+1}(z' | z_0)}{p_{\mathcal{R}}(z', T | z_0, 0)}\mathrm{d}z'$; *(2)* $\mathbf{R}_0^0 \sim q_{traj}^0(\tilde{R}^0), \mathbf{R}_i^0 = \mathbf{R}_{i-1}^T, \forall 0 < i < N$, *then the joint distribution* $q_{\mathcal{R}}(\mathbf{R}_0^0, \mathbf{R}_0^T, \mathbf{R}_1^T, \cdots, \mathbf{R}_{N-1}^T)$ *induced by* $\{(\mathbf{R}_i^t)_{t \in [0, T]}\}_{i \in [N-1]}$ *equals to* $q_{traj}(\tilde{R}^0, ..., \tilde{R}^N)$. *We call this process a chain of equivariant diffusion bridges.*

In this way, a chain of equivariant diffusion bridge can be used to model prior trajectory data, and simulating this chain not only bridges initial and target geometric states but also yields intermediate evolving states. Similarly, we can also use a parameterized model to estimate the scores of bridges in this chain. Instead of having only one objective in all time steps, we now have $N$ bridges in total, which categorize the time span into $N$ groups with different time-dependent objectives. Therefore, by properly specifying time steps and initial conditions, the objective in Eqn. (5) can be seamlessly extended (see Appendix B.7 for more details on its provable guarantee):

$$\mathcal{L}'(\theta) = \mathbb{E}_{(z_0,...,z_N) \sim q_{\mathrm{traj}}(\tilde{R}^0,...,\tilde{R}^N),t,\mathbf{R}_i^{t'}} \lambda(t) \|\mathbf{v}_\theta(\mathbf{R}_i^{t'}, t; z_i) - \nabla_{\mathbf{R}_i^{t'}} \log p_\mathcal{R}^i(z_{i+1}, T|\mathbf{R}_i^{t'}, t')\|^2, \quad (6)$$

where $t \sim \mathcal{U}(0, N \times T), i = \lfloor \frac{t}{T} \rfloor, t' = t - i \times T, \mathbf{R}_i^{t'} \sim q_\mathcal{R}^i(\mathbf{R}_i^{t'}, t'|z_{i+1}, T; z_i, 0)$.

Lastly, we provide the following theoretical result, which further characterizes our framework's expressiveness to completely model the underlying dynamics that induce the trajectory distributions:

**Theorem 3.5.** *Assume* $(\tilde{R}^i)_{i \in [N]}$ *is sampled by simulating a prior SDE on geometric states* $d\tilde{\mathbf{R}}^t = -\nabla H_\mathcal{R}^*(\tilde{\mathbf{R}}^t)dt + \sigma d\tilde{\mathbf{W}}^t$. *Let* $\mu_i^*$ *denote the path measure of this prior SDE when* $t \in [iT, (i+1)T]$. *Building upon* $(\tilde{R}^i)_{i \in [N]}$, *let* $\{\mu_\mathcal{R}^i\}_{i \in [N-1]}$ *denote the path measure of our chain of equivariant diffusion bridges. Under mild assumptions, we have* $\lim_{N \to \infty} \max_i \mathrm{KL}(\mu_i^* || \mu_\mathcal{R}^i) = 0$.

It is noteworthy that the assumption of the prior SDE existence holds in various real-world applications. For example, in geometry optimization, we can formulate the iterative updating process of a molecular system as $d\mathbf{R}^t = -\alpha \nabla_{\mathbf{R}^t} V(\mathbf{R}^t)dt + \beta d\mathbf{W}^t$, where $V(\mathbf{R}^t)$ denotes the potential energy at $\mathbf{R}^t$ and $\alpha, \beta$ are step sizes [88]. From Theorem 3.5, such prior SDE serves as the underlying law governing the evolution dynamic, and our chain of equivariant diffusion bridges constructed from empirical trajectory data can well approximate it, showing the completeness of our framework.

### 3.3 Practical Implementation

In this subsection, we elaborate on how to practically implement our framework. According to Eqn. (5), it is necessary to carefully design (1) tractable distribution $q_\mathcal{R}(\mathbf{R}^t, t|z_1, T; z_0, 0)$ for sampling $\mathbf{R}^t$; (2) closed-form matching objective $\nabla_{\mathbf{R}^t} \log p_\mathcal{R}(z_1, T|\mathbf{R}^t, t)$.

**Matching objective.** Inspired by diffusion models that use Gaussian transition kernels for tractable computation, we design the SDE on geometric states in Proposition 3.1 to be:

$$d\mathbf{R}^t = \sigma d\mathbf{W}^t, \quad \textit{with transition density} \quad p_\mathcal{R}(z', t'|z, t) = \mathcal{N}(z_0, \sigma^2(t'-t)\mathbf{I}) \quad (7)$$

The explicit form of the objective can be directly calculated, i.e., $\nabla_{\mathbf{R}^t} \log p_\mathcal{R}(z_1, T|\mathbf{R}^t, t) = \frac{z_1 - \mathbf{R}^t}{\sigma^2(T-t)}$.

**Sampling distribution.** According to Theorem 3.3, the transition density $q_\mathcal{R}(\mathbf{R}^t, t|z_1, T; z_0, 0)$ can be calculated by using the Doob's $h$-transform in Proposition 3.2, i.e., $q_\mathcal{R}(\mathbf{R}^t, t|z_1, T; z_0, 0) = p_\mathcal{R}(\mathbf{R}^t, t|z_1, T) \frac{h_\mathcal{R}(\mathbf{R}^t, t; z_0)}{h_\mathcal{R}(z_1, T; z_0)}$. Moreover, $h_\mathcal{R}$ is determined by $q_{\mathrm{data}}$ and $p_\mathcal{R}$, which is already specified in Eqn. (7). Therefore, we can also calculate $q_\mathcal{R}(\mathbf{R}^t, t|z_1, T; z_0, 0) = \mathcal{N}(\frac{t}{T} z_1 + \frac{T-t}{T} z_0, \sigma^2 \frac{t(T-t)}{T^2} \mathbf{I})$.

**Symmetry constraints.** In proposition 3.1, we have several conditions that should be satisfied to meet the symmetry constraints. Firstly, since a parameterized model $\mathbf{v}_\theta(\mathbf{R}^t, t; \mathbf{R}^0)$ is used to estimate the score of our equivariant diffusion bridge, it should be SO(3)-equivariant and T(3)-invariant. Besides, we follow [50, 115] to consider CoM-free systems: given $R = \{\mathbf{r}_1, ..., \mathbf{r}_n\}$, we define $\bar{\mathbf{r}} = \frac{1}{n} \sum_{i=1}^n \mathbf{r}_i$ and the CoM-free version of $R = \{\mathbf{r}_1 - \bar{\mathbf{r}}, ..., \mathbf{r}_n - \bar{\mathbf{r}}\}$. To sample from $\mathcal{N}(z_0, \sigma^2 \mathbf{I})$ with $z_0 \in \mathcal{R}$ consisting of $n$ objects, we (1) sample $\boldsymbol{\epsilon} = \{\epsilon_i\}_{i=1}^n$ by i.i.d. drawing $\epsilon_i \sim \mathcal{N}(\mathbf{0}, \mathbf{I}_3)$; (2) calculate the CoM-free $\boldsymbol{\epsilon}'$ of $\boldsymbol{\epsilon}$; (3) obtain $z_0 + \sigma \boldsymbol{\epsilon}'$.

**Trajectory guidance.** According to Eqn. (6), both $p_\mathcal{R}^i$ and $q_\mathcal{R}^i$ for all $i \in [N-1]$ should be determined. Similarly, we set $p_\mathcal{R}^i(z_{i+1}, T|\mathbf{R}^{t'}, t') = \mathcal{N}(\mathbf{R}^{t'}, \sigma_i^2(T-t')\mathbf{I})$, which further induces $q_\mathcal{R}^i(\mathbf{R}^{t'}, t'|z_{i+1}, T; z_i, 0) = \mathcal{N}(\frac{t'}{T} z_{i+1} + \frac{T-t'}{T} z_i, \sigma_i^2 \frac{t'(T-t')}{T^2} \mathbf{I})$.

Combining all the above design choices, we have the following algorithms for training our Geometric Diffusion Bridge (Alg. 3) and leveraging trajectory guidance if available (Alg. 4). After the model

is well trained, we leverage ODE numerical solvers [12] to simulate the bridge process by using its equivalent probability flow ODE [99]. In this way, we can effectively and deterministically predict future geometric states of interest from initial states in an efficient iterative process. Lastly, it is also noteworthy that our framework is general to be implemented by using other advanced design strategies [99, 47, 48], which we leave as future work.

---

**Algorithm 1** Training

1: **repeat**
2: $\quad (z_0, z_1) \sim q_{\text{data}}(R^{t_0}, R^{t_1})$
3: $\quad t \sim \mathcal{U}[0, T]$
4: $\quad \epsilon \sim \mathcal{N}(\mathbf{0}, \mathbf{I})$
5: $\quad \mathbf{R}^t = \frac{t}{T} z_1 + \frac{T-t}{T} z_0 + \frac{\sqrt{t(T-t)}}{T} \sigma \epsilon$
6: $\quad$ Take gradient descent step on
$$\nabla_\theta \lambda(t) \left\| \frac{z_1 - \mathbf{R}^t}{\sigma^2 (T-t)} - \mathbf{v}_\theta(\mathbf{R}^t, t; z_0) \right\|^2$$
7: **until** converged

**Algorithm 2** Training with trajectory guidance

1: **repeat**
2: $\quad (z_0, \ldots, z_N) \sim q_{\text{traj}}(\tilde{R}^0, \ldots, \tilde{R}^N)$
3: $\quad t \sim \mathcal{U}(0, N \times T), i = \lfloor \frac{t}{T} \rfloor, t' = t - i \times T$
4: $\quad \epsilon \sim \mathcal{N}(\mathbf{0}, \mathbf{I})$
5: $\quad \mathbf{R}_i^{t'} = \frac{t'}{T} z_{i+1} + \frac{T-t'}{T} z_i + \frac{\sqrt{t'(T-t')}}{T} \sigma_i \epsilon$
6: $\quad$ Take gradient descent step on
$$\nabla_\theta \lambda(t) \left\| \frac{z_{i+1} - \mathbf{R}_i^{t'}}{\sigma_i^2 (T-t')} - \mathbf{v}_\theta(\mathbf{R}_i^{t'}, t; z_i) \right\|^2$$
7: **until** converged

---

## 4 Experiments

In this section, we empirically study the effectiveness of our Geometric Diffusion Bridge on crucial real-world challenges requiring bridging geometric states. In particular, we carefully design several experiments covering different types of data, scales and scenarios, as shown in Table 2. Due to space limits, we present more details in Appendix D.

Table 2: Summary of experimental setup.

| Dataset | Task Description | Data Type | Trajectory data | Training set size |
|---------|-----------------|-----------|-----------------|-------------------|
| QM9 [79] | Equilibrium State Prediction | Simple molecule | ✗ | 110,000 |
| Molecule3D [116] | Equilibrium State Prediction | Simple molecule | ✗ | 2,339,788 |
| OC22, IS2RS [13] | Structure Relaxation | Adsorbate-Catalyst complex | ✓ | 45,890 |

### 4.1 Equilibrium State Prediction

**Task.** Equilibrium states typically represent local minima on the Born-Oppenheimer potential energy surface of a molecular system [54], which correspond to its most stable geometric state and play an essential role in determining its properties in various aspects [4, 21]. In this task, our goal is to accurately predict the equilibrium state from the initial geometric state of a molecular system.

**Dataset.** Two popular datasets are used: (1) QM9 [79] is a medium-scale dataset that has been widely used for molecular modeling, consisting of $\tilde{1}30,000$ organic molecules. In convention, 110k, 10k, and 11k molecules are used for train/valid/test sets respectively; (2) Molecule3D [116] is a large-scale dataset curated from the PubChemQC project [67, 71], consisting of 3,899,647 molecules in total and its train/valid/test splitting ratio is $6:2:2$. In particular, both random and scaffold splitting methods are adopted to thoroughly evaluate the in-distribution and out-of-distribution performance. For each molecule, an initial geometric state is generated by using fast and coarse force field [73, 52] and geometry optimization is conducted to obtain DFT-calculated equilibrium geometric structure.

**Setting.** In this task, we parameterize $\mathbf{v}_\theta(\mathbf{R}^t, t; \mathbf{R}^0)$ by extending a Graph-Transformer based equivariant network [92, 63] to encode both time steps and initial geometric states as conditions. For inference, we use 10 time steps with the Euler solver [12]. Following [111], we choose several strong baselines for a comprehensive comparison, and use three metrics for measuring the error between predicted target states and ground-truth states: C-RMSD, D-MAE and D-RMSE. The detailed descriptions of the baselines, evaluation metrics and training settings are presented in Appendix D.1.

**Results.** Results on QM9 and Molecule3D are shown in Table 3 and 4 respectively. It can be easily seen that our GDB framework consistently surpasses all baselines by a significantly large margin on

Table 3: Results on the QM9 dataset (Å). We report the official results of baselines from [111]

| | Validation | | | Test | | |
|---|---|---|---|---|---|---|
| | D-MAE↓ | D-RMSE↓ | C-RMSD↓ | D-MAE↓ | D-RMSE↓ | C-RMSD↓ |
| RDKit DG | 0.358 | 0.616 | 0.722 | 0.358 | 0.615 | 0.722 |
| RDKit ETKDG | 0.355 | 0.621 | 0.691 | 0.355 | 0.621 | 0.689 |
| GINE [39] | 0.357 | 0.673 | 0.685 | 0.357 | 0.669 | 0.693 |
| GATv2 [10] | 0.339 | 0.663 | 0.661 | 0.339 | 0.659 | 0.666 |
| GPS [80] | 0.326 | 0.644 | 0.662 | 0.326 | 0.640 | 0.666 |
| GTMGC [111] | 0.262 | 0.468 | 0.362 | 0.264 | 0.470 | 0.367 |
| **GDB (ours)** | **0.092** | **0.218** | **0.143** | **0.096** | **0.223** | **0.148** |

Table 4: Results on the Molecule3D dataset (Å). We report the official results of baselines from [111]

| | Validation | | | Test | | |
|---|---|---|---|---|---|---|
| | D-MAE↓ | D-RMSE↓ | C-RMSD↓ | D-MAE↓ | D-RMSE↓ | C-RMSD↓ |
| (a) Random Split | | | | | | |
| RDKit DG | 0.581 | 0.930 | 1.054 | 0.582 | 0.932 | 1.055 |
| RDKit ETKDG | 0.575 | 0.941 | 0.998 | 0.576 | 0.942 | 0.999 |
| DeeperGCN-DAGNN [116] | 0.509 | 0.849 | * | 0.571 | 0.961 | * |
| GINE [39] | 0.590 | 1.014 | 1.116 | 0.592 | 1.018 | 1.116 |
| GATv2 [10] | 0.563 | 0.983 | 1.082 | 0.564 | 0.986 | 1.083 |
| GPS [80] | 0.528 | 0.909 | 1.036 | 0.529 | 0.911 | 1.038 |
| GTMGC [111] | 0.432 | 0.719 | 0.712 | 0.433 | 0.721 | 0.713 |
| **GDB (ours)** | **0.374** | **0.631** | **0.622** | **0.376** | **0.626** | **0.619** |
| (b) Scaffold Split | | | | | | |
| RDKit DG | 0.542 | 0.872 | 1.001 | 0.524 | 0.857 | 0.973 |
| RDKit ETKDG | 0.531 | 0.874 | 0.928 | 0.511 | 0.859 | 0.898 |
| DeeperGCN-DAGNN [116] | 0.617 | 0.930 | * | 0.763 | 1.176 | * |
| GINE [39] | 0.883 | 1.517 | 1.407 | 1.400 | 2.224 | 1.960 |
| GATv2 [10] | 0.778 | 1.385 | 1.254 | 1.238 | 2.069 | 1.752 |
| GPS [80] | 0.538 | 0.885 | 1.031 | 0.657 | 1.091 | 1.136 |
| GTMGC [111] | 0.406 | 0.675 | 0.678 | 0.400 | 0.679 | 0.693 |
| **GDB (ours)** | **0.335** | **0.587** | **0.592** | **0.341** | **0.608** | **0.603** |

QM9, e.g., 60.5%/59.7% relative C-RMSD reduction on valid/test sets respectively, establishing a new state-of-the-art performance. Similar trends also can be observed in Molecule3D, i.e., 12.6%/13.2% relative C-RMSD reduction for valid/test sets of the random split and 12.7%/13.0% reduction for the scaffold split, largely outperforming the best baseline. These significant error reduction results show the superiority of our GDB framework for bridging geometric states, and its generality on both medium and large-scale challenges. Moreover, our framework performs consistently across valid and tests of both random and scaffold splits, further verifying its robustness in challenging scenarios.

## 4.2 Structure Relaxation

**Task.** Catalyst discovery is crucial for various applications. Adsorbate candidates are placed on catalyst surfaces and evolve through structure relaxation to adsorption states, in which the adsorption structures can be determined for measuring catalyst activity and selectivity. Our goal is thus to accurately predict adsorption states from initial states of adsorbate-catalyst complexes.

**Dataset.** We adopt Open Catalyst 2022 (OC22) dataset [105], which has great significance for the development of Oxygen Evolution Reaction (OER) catalysts. Each data is in the form of the adsorbate-catalyst complex. Both initial and adsorption states with trajectories connecting them are provided. The training set consists of 45,890 catalyst-adsorbate complexes. To better evaluate the model's performance, the validation and test sets consider the in-distribution (ID) and out-of-distribution (OOD) settings which use unseen catalysts, containing approximately 2,624 and 2,780 complexes respectively.

**Setting.** Following [105], we use the Average Distance within Threshold (ADwT) as the evaluation metric, which reflects the percentage of structures with an atom position MAE below thresholds. We parameterize $\mathbf{v}_\theta(\mathbf{R}^t, t; \mathbf{R}^0)$ by using GemNet-OC [34], which also serves as a verification that

Table 5: Results on the OC22 IS2RS Validation set. "OC20+OC22" denotes using both OC20 [13] and OC22 data; "OC20→OC22" means pre-training on OC20 data then fine-tuning on OC22 data; "OC22-only" means only using OC22 data. We report the official results of baselines from [105]

| Model | ADwT [%] ↑ (ID) | ADwT [%] ↑ (OOD) | Avg [%] ↑ |
|---|---|---|---|
| OC20+OC22 | | | |
| SpinConv [94] | 55.79 | 47.31 | 51.55 |
| GemNet-OC [34] | 60.99 | 53.85 | 57.42 |
| OC20→OC22 | | | |
| SpinConv [94] | 56.69 | 45.78 | 51.23 |
| GemNet-OC [34] | 58.03 | 48.33 | 53.18 |
| GemNet-OC-Large [34] | 59.69 | 51.66 | 55.67 |
| OC22-only | | | |
| IS baseline | 44.77 | 42.59 | 43.68 |
| SpinConv [94] | 54.53 | 40.45 | 47.49 |
| GemNet-dT [32] | 59.68 | 51.25 | 55.46 |
| GemNet-OC [34] | 60.69 | 52.90 | 56.79 |
| **GDB (ours)** | **63.01** | **55.78** | **59.39** |
| − trajectory guidance | 62.14 | 54.94 | 58.54 |
| − $\mathbf{R}^0$ condition | 60.17 | 49.26 | 54.71 |

our framework is compatible with different backbone models. For inference, we also use 10 time steps with the Euler solver. Following [105], we choose strong MLFF baselines trained on force field data for a challenging comparison. The detailed descriptions of baselines and settings are presented in Appendix D.2.

**Results.** In Table 5, our GDB significantly outperforms the best baseline, e.g., 3.3%/3.6%/3.4% relative improvement on the ADwT metric of ID, OOD and Avg respectively. It is noteworthy that the best baseline is the GemNet-OC force field trained on both OC20 and OC22 data, which is 10 times more than OC22 data only. Nevertheless, our framework still achieves better performance on predicting the adsorption geometric states. Moreover, our framework without using any trajectory data still can achieve better performance compared to the best baseline, e.g., 58.54 v.s. 57.42 Avg[%]. All the results on this challenging task further demonstrate the superiority and completeness of our framework.

**Ablation study.** Furthermore, we conduct ablation studies to examine key designs of our framework in Table 5. Firstly, we can see that using trajectory guidance indeed improves the performance of our framework, e.g., 1.4% relative improvement on Avg ADwT. Moreover, we also investigate the impact of $\mathbf{R}^0$ condition in $\mathbf{v}_\theta(\mathbf{R}^t, t; \mathbf{R}^0)$, which plays an essential role in preserving the joint distribution of geometric states. Without this condition, we can see a significant drop, e.g., 6.5%/10.3% relative ADwT drop on Avg/OOD respectively. Overall, these ablation studies serve as strong supports on the necessity of developing a unified framework that can precisely bridge geometric states by preserving their joint distributions and effectively leverage trajectory data as guidance for enhanced performance.

## 5 Related Works

**Direct Prediction.** One line of approach for bridging geometric states is direct prediction, i.e., training a model to directly predict target geometric states given initial states as input. Models that carefully respect symmetry constraints such as the equivariance to 3D rotations and translations are typically used, which are called Geometric Equivariant Networks [11, 36, 120, 27]. Different techniques have been explored to encode such priors, which mainly include vector operations such as scalar and vector product [35, 87, 89, 41, 103, 14], e.g., the scalar-vector product used in EGNN [87], and tensor product based operations [104, 31, 8, 57, 64]. Despite its simplicity and efficiency, direct prediction requires encoding the iterative evolution of geometric states into a single-step prediction model, which lacks the ability to capture the underlying dynamics and cannot leverage trajectories of geometric states.

**Machine Learning Force Field.** Another line of approach is called machine learning force field (MLFF) [106, 5, 6, 70, 75, 58], which are trained to predict intermediate labels, such as the potential

energy or force of the (local) current geometric state instead. After training, MLFFs can be used to simulate the trajectory of geometric states over time based on underlying equations. Using Geometric Equivariant Networks as the backbone, MLFFs typically satisfy the symmetry constraints. Besides, trajectory data with additional energy or force labels can directly be used for training MLFFs. However, this paradigm highly depends on the existence and quality of intermediate labels since small local errors in energy or force prediction can accumulate along the simulation process [7, 106, 30]. Moreover, there exists no guarantee that MLFFs can completely model joint state distributions, which is another limitation for bridging geometric states.

**Geometric Diffusion Models.** In recent years, diffusion models [37, 99] have emerged with state-of-the-art generative modeling performance across various domains [85, 108, 51, 56]. In geometric domain, diffusion models are typically used for molecule conformation generation [115, 114, 38] and protein design [108, 117]. By properly design the noising process and model architectures, symmetry constraints on the transition kernel and prior distribution can be satisfied, which guarantees the generated data is sampled from roto-translational invariant distributions [115, 38]. In addition to the score-based formulation, recent advances further extend new techniques such as flow matching [59, 61, 1] to satisfy symmetry constraints for these generation tasks [49, 100]. Nevertheless, there exists no guaratee that these approaches can model the joint distribution of geometric states [61, 96]. And how to leverage trajectory data as guidance for bridging geometric states is also challenging.

**Other techniques.** MoreRed [45] trains a diffusion model on equilibrium molecule conformations with a time step predictor, and directly use it for bridging any conformations to their equilibrium states. GTMGC [111] instead develop a Graph Transformer to directly predict equilibrium conformations from their 2D graph forms. Both of them are limited to the equilibrium conformation prediction task, cannot preserve the joint state distribution and leverage trajectory data. EGNO [112] is a concurrent work that develops a neural operator based approach to model dynamics of trajectories. By carefully designing temporal convolution in fourier spaces, EGNO can learn from trajectory data. However, this tailored approach cannot be directly used without trajectory guidance. To preserve joint data distributions, [22, 121] coincide with us to leverage Doob's $h$-transform to repurposing standard diffusion processes, but they do not respect symmetry constraints and cannot leverage trajectories. There also exist recent works that study the diffusion bridge framework [76, 93] and apply it to various domains such as images and graphs [110, 62, 42]. Compared to all above approaches, our GDB framework stands out as a unique and ideal solution that can precisely bridge geometric states and effectively leverage trajectory data (if available) in a unified manner.

# 6   Conclusion

In this work, we introduce Geometric Diffusion Bridge (GDB), a general framework for bridging geometric states through generative modeling. We leverage a modified version of Doob's $h$-transform to constructe an equivariant diffusion bridge for bridging initial and target geometric states. Trajectory data can further be seamlessly leveraged as guidance by using a chain of equivariant diffusion bridges, allowing complete modeling of trajectory data. Mathematically, we conduct a comprehensive theoretical analysis showing our framework's ability to preserve joint distributions of geometric states and capability to completely model the evolution dynamics. Empirical comparisons on different settings show that our GDB significantly surpasses existing state-of-the-art approaches and ablation studies further underscore the necessity of several key designs in our framework. In the future, it is worth exploring better implementation strategies of our framework for enhanced performance, and applying our GDB to other critical challenges involving bringing geometric states.

## Broader Impacts and Limitations

This work newly proposes a general framework to bridge geometric states, which has great significance in various scientific domains. Our experimental results have also demonstrated considerable positive potential for various applications, such as catalyst discovery and molecule optimization, which can significantly contribute to the advancement of renewable energy processes and chemistry discovery. However, it is essential to acknowledge the potential negative impacts including the development of toxic drugs and materials. Thus, stringent measures should be implemented to mitigate these risks.

There also exist some limitations to our work. For the sake of generality, we do not experiment with advanced implementation strategies of training objectives and sampling algorithms, which leave room for further improvement. Besides, the employment of Transformer-based architectures may also limit the efficiency of our framework. This has also become a common issue in transformer-based diffusion models, which we have earmarked for future research.

## Acknowledgements

We thank all the anonymous reviewers for the very careful and detailed reviews as well as the valuable suggestions. Their help has further enhanced our work. Liwei Wang is supported by National Science and Technology Major Project (2022ZD0114902) and National Science Foundation of China (NSFC62276005). Di He is supported by National Science Foundation of China (NSFC62376007).

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

# A  Organization of the Appendix

The supplementary material is organized as follows. In Appendix B, we first recall some definitions and tools from stochastic calculus and then give the proofs of all theorems. In Appendix C, we give the derivation of our practical objective function and our sampling algorithms. In Appendix D, we give some details of our experiments, including a comprehensive introduction to the datasets, baselines, metrics and settings.

# B  Proof of Theorems

## B.1  Review of Stochastic Calculus

Let $(X_t)_{t \in [0,T]}$ be a stochastic process. We use $p(x', t'|x_1, t_1; x_2, t_2; \ldots; x_n, t_n)$ to denote its conditional density function satisfying

$$P(\mathbf{X}_{t'} \in A | \mathbf{X}_{t_1} = x_1, \mathbf{X}_{t_2} = x_2, \ldots, \mathbf{X}_{t_n} = x_n) = \int_A p(x', t'|x_1, t_1; x_2, t_2; \ldots; x_n, t_n) \mathrm{d}x'$$

for any Borel set $A$, where $t_1 < t_2 < \cdots < t_n$. If $(\mathbf{X}_t)_{t \in [0,T]}$ is a Markov process, $p(x', t'|x_1, t_1; x_2, t_2; \ldots; x_n, t_n) = p(x', t'|x_n, t_n)$, which is also called a transition density function.

One of the most important results of stochastic calculus is the Ito's formula. The precise statements are as follows.

**Theorem B.1** (Ito's formula for Brownian Motion). *Let $\mathbf{B}_t$ be the $d-$dimensional Brownian Motion. Assume $f$ is a bounded real valued function with continuous second-order partial derivatives, i.e. $f \in C_b^2(\mathbb{R}^d)$. Then the Ito's formula is given by*

$$f(\mathbf{B}_t) = f(\mathbf{B}_0) + \int_0^T \nabla f(\mathbf{B}_t) \cdot \mathrm{d}\mathbf{B}_t + \frac{1}{2} \int_0^T \nabla^2 f(\mathbf{B}_t) \mathrm{d}t. \tag{8}$$

We follow [86] for the proof of Doob's h-transform. The infinitesimal generator of the Markov process plays an important role in the proof of the Doob's h-transform. The precise definitions are as follows.

**Definition B.2.** *(Generator of a Process) The infinitesimal generator $\mathcal{A}_t$ of a stochastic process $(\mathbf{X}_t)$ for a function $\phi(x)$ is*

$$\mathcal{A}_t \phi(x) = \lim_{s \to 0^+} \frac{\mathbb{E}[\phi(\mathbf{X}_{t+s})|\mathbf{X}_t = x] - \phi(x)}{s}, \tag{9}$$

*where $\phi$ is a suitably regular function. For an Itô process defined as the solution to the SDE $\mathrm{d}\mathbf{X}_t = \mathbf{f}(\mathbf{X}_t, t)\mathrm{d}t + \sigma(t)\mathrm{d}\mathbf{B}_t$, the generator is*

$$\mathcal{A}_t = \sum_{i=1}^d \mathbf{f}^i(x, t) \frac{\partial}{\partial x_i} + \frac{1}{2} \sum_{i=1}^d \sigma^2(t) \frac{\partial^2}{\partial x_i^2}. \tag{10}$$

The Fokker-Planck's Equation is an useful tool to track the evolution of the transition density function associated with an SDE. The precise statements are as follows.

**Proposition B.3.** *(Fokker-Planck's Equation) Let $p(x', t'|x, t)$ be the transition density function of the SDE $\mathrm{d}\mathbf{X}_t = \mathbf{f}(\mathbf{X}_t, t)\mathrm{d}t + \sigma(t)\mathrm{d}\mathbf{B}_t$. Then $p(x', t'|x, t)$ satisfies the Fokker-Planck's Equation*

$$\frac{\partial p(x, t|x_0, 0)}{\partial t} = -\sum_{i=1}^d \frac{\partial (\mathbf{f}^i(x, t)p(x, t|x_0, 0))}{\partial x_i} + \frac{1}{2} \sum_{i=1}^d \sigma^2(t) \frac{\partial^2 p(x, t|x_0, 0)}{\partial x_i^2} = 0, \tag{11}$$

*with the initial condition $p(x, 0|x_0, 0) = \delta(x - x_0)$. The Fokker-Planck's Equation can also be written in a compact form using the generator $\mathcal{A}_t$:*

$$\frac{\partial}{\partial t} p(x, t|x_0, 0) = \mathcal{A}_t^* p(x, t|x_0, 0), \tag{12}$$

*where $\mathcal{A}_t^*$ is the adjoint operator of $\mathcal{A}$:*

$$\mathcal{A}_t^* = -\sum_{i=1}^d \frac{\partial (\mathbf{f}^i(x, t) \cdot)}{\partial x_i} + \frac{1}{2} \sum_{i=1}^d \sigma^2(t) \frac{\partial^2 (\cdot)}{\partial x_i^2}. \tag{13}$$

When the terminal is fixed, the evolution of the transition density function can also given by a PDE, which is called the Backward Kolmogorov Equation. We give the precise statement as follows.

**Proposition B.4.** *(Backward Kolmogorov Equation) Let $p(x', t'|x, t)$ be the transition density function of the SDE $\mathrm{d}\mathbf{X}_t = \mathbf{f}(\mathbf{X}_t, t)\mathrm{d}t + \sigma(t)\mathrm{d}\mathbf{B}_t$. Then $p(x', t'|x, t)$ satisfies the Backward Kolmogorov Equation*

$$-\frac{\partial p(x_t, t|x, s)}{\partial s} = \sum_{i=1}^{d} \mathbf{f}^i(x, s)\frac{\partial p(x_t, t|x, s)}{\partial x_i} + \frac{1}{2}\sum_{i=1}^{d}\sigma^2(s)\frac{\partial^2 p(x_t, t|x, s)}{\partial x_i^2} = 0, \qquad (14)$$

*with the initial condition $p(x_t, t|x, t) = \delta(x - x_t)$. The Backward Kolmogorov Equation can also be written in a compact form using the generator $\mathcal{A}_s$:*

$$\left(\frac{\partial}{\partial s} + \mathcal{A}_s\right) p(x_t, t|x, s) = 0. \qquad (15)$$

## B.2   Proof of Proposition 3.1

**Proposition B.5.** *Let $\mathcal{R}$ denote the space of geometric states and $\mathbf{f}_\mathcal{R}(\cdot, \cdot) : \mathcal{R} \times [0, T] \to \mathcal{R}$ denote the drift coefficient on $\mathcal{R}$. Let $(\mathbf{W}^t)_{t \in [0,T]}$ denote the Wiener process on $\mathcal{R}$. Given an SDE on geometric states $\mathrm{d}\mathbf{R}^t = \mathbf{f}_\mathcal{R}(\mathbf{R}^t, t)\mathrm{d}t + \sigma(t)\mathrm{d}\mathbf{W}^t$, $\mathbf{R}^0 \sim q(\mathbf{R}^0)$, its transition density $p_\mathcal{R}(z', t'|z, t), z, z' \in \mathcal{R}$ is SE(3)-equivariant, i.e., $p_\mathcal{R}(\mathbf{R}^{t'}, t'|\mathbf{R}^t, t) = p_\mathcal{R}(\rho^\mathcal{R}(g)[\mathbf{R}^{t'}], t'|\rho^\mathcal{R}(g)[\mathbf{R}^t], t), \forall g \in \mathrm{SE}(3), \forall 0 \le t < t' \le T$, if the following conditions are satisfied: (1) $q(\mathbf{R}^0)$ is SE(3)-invariant; (2) $\mathbf{f}_\mathcal{R}(\cdot, t)$ is SO(3)-equivariant and T(3)-invariant; (3) the transition density of $(\mathbf{W}^t)_{t \in [0,T]}$ is SE(3)-equivariant.*

*Proof.* In this section, we view $R = \{\mathbf{r}_1, ..., \mathbf{r}_n\} \in \mathcal{R}$ as $\mathbf{r}_1 \oplus \mathbf{r}_2 \oplus \cdots \mathbf{r}_n \in \mathbb{R}^{3n}$, which is the concatenation of $\mathbf{r}_i$. So from this perspective, the space $\mathcal{R}$ is isomorphic to the Euclidean space $\mathbb{R}^{3n}$. Then $(\mathbf{W}^t)_{t \in [0,T]}$ is the Wiener process with dimension $d = 3n$.

For any $g \in \mathrm{SE}(3)$, $\rho^\mathcal{R}(g)$ can be characterized by an orthogonal matrix $\mathbf{O}(g) \in \mathbb{R}^{3 \times 3}$, satisfying $\det(\mathbf{O}(g)) = 1$, and a translation vector $\mathbf{t} \in \mathbb{R}^3$. Then the representation of SE(3) on $\mathbb{R}^{3n}$ is given by

$$\rho^\mathcal{R}(g)[R] = \mathbf{O}_\mathcal{R}(g)R + \mathbf{t}_\mathcal{R}, \qquad (16)$$

where $\mathbf{O}_\mathcal{R}(g) = \mathrm{diag}\{\mathbf{O}(g), \mathbf{O}(g), \ldots, \mathbf{O}(g)\}$, $\mathbf{t}_\mathcal{R} = \mathbf{t} \oplus \mathbf{t} \oplus \cdots \mathbf{t} \in \mathbb{R}^{3n}$. It's obvious that $\mathbf{O}_\mathcal{R}(g)$ is also an orthogonal matrix in $\mathbb{R}^{3n \times 3n}$, satisfying $\mathbf{O}_\mathcal{R}^{-1}(g) = \mathbf{O}_\mathcal{R}^T(g)$.

According to Proposition B.3, the evolution of the transition density function is given by the Fokker-Planck's Equation

$$\frac{\partial p_\mathcal{R}(x, t|x_0, 0)}{\partial t} = -\sum_{i=1}^{d}\frac{\partial\left(\mathbf{f}^i(x, t)p_\mathcal{R}(x, t|x_0, 0)\right)}{\partial x_i} + \frac{1}{2}\sum_{i=1}^{d}\sigma^2(t)\frac{\partial^2\left(p_\mathcal{R}(x, t|x_0, 0)\right)}{\partial x_i^2}, \qquad (17)$$

with the initial condition $p_\mathcal{R}(x, 0|x_0, 0) = \delta(x - x_0)$.

Let $y = \mathbf{O}_\mathcal{R}(g)x + \mathbf{t}_\mathcal{R}$, $y_0 = \mathbf{O}_\mathcal{R}(g)x_0 + \mathbf{t}_\mathcal{R}$, then we have

$$p_\mathcal{R}(\rho^\mathcal{R}(g)[x], t|\rho^\mathcal{R}(g)[x_0], 0) = p_\mathcal{R}(\mathbf{O}_\mathcal{R}(g)x + \mathbf{t}_\mathcal{R}, t|\mathbf{O}_\mathcal{R}(g)x_0 + \mathbf{t}_\mathcal{R}, 0) = p_\mathcal{R}(y, t|y_0, 0). \qquad (18)$$

The evolution of the transition density function $p_\mathcal{R}(y, t|y_0, 0)$ is also given by the Fokker-Planck's Equation:

$$\frac{\partial p_\mathcal{R}(y, t|y_0, 0)}{\partial t} = -\sum_{i=1}^{d}\frac{\partial\left(\mathbf{f}^i(y, t)p_\mathcal{R}(y, t|y_0, 0)\right)}{\partial y_i} + \frac{1}{2}\sum_{i=1}^{d}\sigma^2(t)\frac{\partial^2\left(p_\mathcal{R}(y, t|y_0, 0)\right)}{\partial y_i^2}, \qquad (19)$$

with the boundary condition $p_\mathcal{R}(y, 0|y_0, 0) = \delta(y - y_0) = \delta(x - x_0)$. Since $y = \mathbf{O}_\mathcal{R}(g)x + \mathbf{t}_\mathcal{R}$, we have $x = \mathbf{O}_\mathcal{R}^{-1}(g)(y - \mathbf{t}_\mathcal{R})$. Then by the chain rule, we have

$$\frac{\partial}{\partial y_i} = \sum_{j=1}^{d}\frac{\partial x_j}{\partial y_i}\frac{\partial}{\partial x_j} = \sum_{j=1}^{d}(\mathbf{O}_\mathcal{R}^{-1}(g))_{ji}\frac{\partial}{\partial x_j} = \sum_{j=1}^{d}(\mathbf{O}_\mathcal{R}(g))_{ij}\frac{\partial}{\partial x_j}. \qquad (20)$$

Since $\mathbf{f}_{\mathcal{R}}(\cdot, t)$ is a SO(3)-equivariant and T(3)-invariant function, we have

$$\mathbf{f}_{\mathcal{R}}^i(y, t) = \mathbf{f}_{\mathcal{R}}^i(\mathbf{O}_{\mathcal{R}}(g)x + \mathbf{t}_{\mathcal{R}}, t) = (\mathbf{O}_{\mathcal{R}}(g)\mathbf{f}_{\mathcal{R}}(x, t))_i = \sum_{k=1}^d (\mathbf{O}_{\mathcal{R}}(g))_{ik}\mathbf{f}_{\mathcal{R}}^k(x, t). \tag{21}$$

Then the Fokker-Planck's equation becomes

$$\frac{\partial p_{\mathcal{R}}(y, t|y_0, 0)}{\partial t} = -\sum_{i=1}^d \frac{\partial\left(\mathbf{f}^i(y, t)p_{\mathcal{R}}(y, t|y_0, 0)\right)}{\partial y_i} + \frac{1}{2}\sigma^2(t)\sum_{i=1}^d \frac{\partial^2\left(p_{\mathcal{R}}(y, t|y_0, 0)\right)}{\partial y_i^2} \tag{22}$$

$$= -\sum_{i=1}^d\sum_{j=1}^d\sum_{k=1}^d (\mathbf{O}_{\mathcal{R}}(g))_{ij}\frac{\partial((\mathbf{O}_{\mathcal{R}}(g))_{ik}\mathbf{f}_{\mathcal{R}}^k(x, t)p_{\mathcal{R}}(y, t|y_0, 0))}{\partial x_j} \tag{23}$$

$$+ \frac{1}{2}\sigma^2(t)\sum_{i=1}^d\sum_{j=1}^d\sum_{k=1}^d (\mathbf{O}_{\mathcal{R}}(g))_{ik}\frac{\partial}{\partial x_k}(\mathbf{O}_{\mathcal{R}}(g))_{ij}\frac{\partial\left(p_{\mathcal{R}}(y, t|y_0, 0)\right)}{\partial x_j} \tag{24}$$

$$= -\sum_{i=1}^d\sum_{j=1}^d\sum_{k=1}^d (\mathbf{O}_{\mathcal{R}}(g))_{ij}(\mathbf{O}_{\mathcal{R}}(g))_{ik}\frac{\partial(\mathbf{f}_{\mathcal{R}}^k(x, t)p_{\mathcal{R}}(y, t|y_0, 0))}{\partial x_j} \tag{25}$$

$$+ \frac{1}{2}\sigma^2(t)\sum_{i=1}^d\sum_{j=1}^d\sum_{k=1}^d (\mathbf{O}_{\mathcal{R}}(g))_{ik}(\mathbf{O}_{\mathcal{R}}(g))_{ij}\frac{\partial}{\partial x_k}\frac{\partial\left(p_{\mathcal{R}}(y, t|y_0, 0)\right)}{\partial x_j}. \tag{26}$$

Since $\mathbf{O}_{\mathcal{R}}(g)$ is an orthogonal matrix, the columns of $\mathbf{O}_{\mathcal{R}}(g)$ are orthogonal to each other, i.e.

$$\sum_{i=1}^d (\mathbf{O}_{\mathcal{R}}(g))_{ik}(\mathbf{O}_{\mathcal{R}}(g))_{ij} = \delta_{jk} = \begin{cases} 0 & j \neq k, \\ 1 & j = k. \end{cases} \tag{27}$$

So the Fokker-Planck's equation can be simplified to

$$\frac{\partial p_{\mathcal{R}}(y, t|y_0, 0)}{\partial t} = -\sum_{j=1}^d\sum_{k=1}^d \frac{\partial(\mathbf{f}_{\mathcal{R}}^k(x, t)p_{\mathcal{R}}(y, t|y_0, 0))}{\partial x_j} \tag{28}$$

$$+ \frac{1}{2}\sigma^2(t)\sum_{j=1}^d\sum_{k=1}^d \delta_{jk}\frac{\partial}{\partial x_k}\frac{\partial\left(p_{\mathcal{R}}(y, t|y_0, 0)\right)}{\partial x_j} \tag{29}$$

$$= -\sum_{j=1}^d \frac{\partial(\mathbf{f}_{\mathcal{R}}^j(x, t)p_{\mathcal{R}}(y, t|y_0, 0))}{\partial x_j} + \frac{1}{2}\sigma^2(t)\sum_{j=1}^d \frac{\partial^2\left(p_{\mathcal{R}}(y, t|y_0, 0)\right)}{\partial(x_j)^2}, \tag{30}$$

which is same as Eqn.(17). Since the boundary condition $p_{\mathcal{R}}(y, 0|y_0, t_0) = \delta(y - y_0) = \delta(x - x_0) = p_{\mathcal{R}}(x, 0|x_0, 0)$, then $p_{\mathcal{R}}(y, t|y_0, t_0) = p_{\mathcal{R}}(x, t|x_0, t_0), \forall t \in [0, T]$. Thus we have proved that $p_{\mathcal{R}}(\mathbf{R}^{t'}, t'|\mathbf{R}^t, t) = p_{\mathcal{R}}(\rho^{\mathcal{R}}(g)[\mathbf{R}^{t'}], t'|\rho^{\mathcal{R}}(g)[\mathbf{R}^t], t), \forall g \in \text{SE}(3), \forall 0 \leq t < t' \leq T$. $\qquad\square$

### B.3 Proof of Proposition 3.2

**Proposition B.6** (Doob's $h$-transform). *Let $p_{\mathcal{R}}(z', t'|z, t)$ be the transition density of the SDE in Proposition 3.1. Let $h_{\mathcal{R}}(\cdot, \cdot) : \mathcal{R} \times [0, T] \to \mathbb{R}_{>0}$ be a smooth function satisfying: (1) $h_{\mathcal{R}}(\cdot, t)$ is SE(3)-invariant; (2) $h_{\mathcal{R}}(z, t) = \int p_{\mathcal{R}}(z', t'|z, t)h_{\mathcal{R}}(z', t')dz'$. We can derive the following $h_{\mathcal{R}}$-transformed SDE on geometric states:*

$$d\mathbf{R}^t = \left[\mathbf{f}_{\mathcal{R}}(\mathbf{R}^t, t) + \sigma^2(t)\nabla_{\mathbf{R}^t}\log h_{\mathcal{R}}(\mathbf{R}^t, t)\right]dt + \sigma(t)d\mathbf{W}_t, \tag{31}$$

*with transition density $p_{\mathcal{R}}^h(z', t'|z, t) = p_{\mathcal{R}}(z', t'|z, t)\frac{h_{\mathcal{R}}(z', t')}{h_{\mathcal{R}}(z, t)}$ preserving the symmetry constraints.*

*Proof.* We use the definition of the infinitesimal generator to prove the proposition. The infinitesimal generator of $p_{\mathcal{R}}^h(x', t'|x, t)$ for a function $\phi(x)$ is given by

$$\mathcal{A}_t^h\phi(x) = \lim_{s \to 0^+} \frac{\mathbb{E}^h[\phi(\mathbf{R}^{t+s})|\mathbf{R}^t = x] - \phi(x)}{s}. \tag{32}$$

Since $p_{\mathcal{R}}^h(z', t'|z, t) = p_{\mathcal{R}}(z', t'|z, t) \frac{h_{\mathcal{R}}(z', t')}{h_{\mathcal{R}}(z, t)}$, so we have

$$\mathbb{E}^h[\phi(\mathbf{R}^{t+s})|\mathbf{R}^t = x] = \frac{\mathbb{E}[\phi(\mathbf{R}^{t+s})h(\mathbf{R}^{t+s}, t+s)|\mathbf{R}^t = x]}{h(x, t)}. \tag{33}$$

Then $\mathcal{A}_t^h \phi(x)$ can be simplified as

$$\mathcal{A}_t^h \phi(x) = \lim_{s \to 0^+} \frac{\mathbb{E}^h[\phi(\mathbf{R}^{t+s})|\mathbf{R}^t = x] - \phi(x)}{s} \tag{34}$$

$$= \lim_{s \to 0^+} \frac{\mathbb{E}[\phi(\mathbf{R}^{t+s})h(\mathbf{R}^{t+s}, t+s)|\mathbf{R}^t = x] - \phi(x)h(x, t)}{sh(x, t)} \tag{35}$$

$$= \frac{1}{h(x, t)}[\frac{\partial h(x, t)}{\partial t}\phi(x) + \sum_{i=1}^d \left( \frac{\partial h(x, t)}{\partial x_i}\phi(x) + h(x, t)\frac{\partial \phi(x)}{\partial x_i} \right) \mathbf{f}^i(x, t) \tag{36}$$

$$+ \frac{1}{2}\sum_{i=1}^d \sigma^2(t)\frac{\partial^2 h(x, t)}{\partial x_i^2}\phi(x) + \sum_{i=1}^d \sigma^2(t)\frac{\partial h(x, t)}{\partial x_i}\frac{\partial \phi(x)}{\partial x_i} \tag{37}$$

$$+ \frac{1}{2}\sum_{i=1}^d \sigma^2(t)\frac{\partial^2 \phi(x)}{\partial x_i^2}h(x, t)] \tag{38}$$

$$= \frac{1}{h(x, t)}[\frac{\partial h(x, t)}{\partial t}\phi(x) + (\mathcal{A}_t h(x, t))\,\phi(x) \sum_{i=1}^d h(x, t)\frac{\partial \phi(x)}{\partial x_i}\mathbf{f}^i(x, t) \tag{39}$$

$$+ \sum_{i=1}^d \sigma^2(t)\frac{\partial h(x, t)}{\partial x_i}\frac{\partial \phi(x)}{\partial x_i} + \frac{1}{2}\sum_{i=1}^d \sigma^2(t)\frac{\partial^2 \phi(x)}{\partial x_i^2}h(x, t)] \tag{40}$$

Since $h(x, t) = \int p_{\mathcal{R}}(x', t'|x, t)h(x', t')\mathrm{d}x$, we have

$$\left( \frac{\partial}{\partial t} + \mathcal{A}_t \right) h(x, t) = \int \left( \frac{\partial p(x', t'|x, t)}{\partial t} + \mathcal{A}_t p(x', t'|x, t) \right) h(x', t')\mathrm{d}x. \tag{41}$$

According to the Backward Kolmogorov Equation (Proposition B.4), we get

$$\frac{\partial p(x', t'|x, t)}{\partial t} + \mathcal{A}_t p(x', t'|x, t) = 0. \tag{42}$$

So we get

$$\left( \frac{\partial}{\partial t} + \mathcal{A}_t \right) h(x, t) = 0. \tag{43}$$

Then $\mathcal{A}_t^h \phi(x)$ can be simplified as

$$\mathcal{A}^h \phi(x) = \frac{1}{h(x, t)}[\sum_{i=1}^d h(x, t)\frac{\partial \phi(x)}{\partial x_i}\mathbf{f}^i(x, t) + \sum_{i=1}^d \sigma^2(t)\frac{\partial h(x, t)}{\partial x_i}\frac{\partial \phi(x)}{\partial x_i} \tag{44}$$

$$+ \frac{1}{2}\sum_{i=1}^d \sigma^2(t)\frac{\partial^2 \phi(x)}{\partial x_i^2}h(x, t)] \tag{45}$$

$$= \sum_{i=1}^d \frac{\partial \phi(x)}{\partial x_i}\mathbf{f}^i(x, t) + \sum_{i=1}^d \sigma^2(t)\frac{1}{h(x, t)}\frac{\partial h(x, t)}{\partial x_i}\frac{\partial \phi(x)}{\partial x_i} + \frac{1}{2}\sum_{i=1}^d \sigma^2(t)\frac{\partial^2 \phi(x)}{\partial x_i^2} \tag{46}$$

$$= \sum_{i=1}^d \left( \mathbf{f}^i(x, t) + \sigma^2(t)\frac{\partial \log h(x, t)}{\partial x_i} \right) \frac{\partial \phi(x)}{\partial x_i} + \frac{1}{2}\sum_{i=1}^d \sigma^2(t)\frac{\partial^2 \phi(x)}{\partial x_i^2}. \tag{47}$$

So we show that

$$\mathcal{A}_t^h = \sum_{i=1}^d \left( \mathbf{f}^i(x, t) + \sigma^2(t)\frac{\partial \log h(x, t)}{\partial x_i} \right) \frac{\partial}{\partial x_i} + \frac{1}{2}\sum_{i=1}^d \sigma^2(t)\frac{\partial^2}{\partial x_i^2}. \tag{48}$$

According to the correspondence between SDE and its generator (Definition B.2), the equation above implies that the h-transformed SDE is given by

$$d\mathbf{R}^t = \left[\mathbf{f}_{\mathcal{R}}(\mathbf{R}^t, t) + \sigma^2(t)\nabla_{\mathbf{R}^t} \log h_{\mathcal{R}}(\mathbf{R}^t, t)\right] dt + \sigma(t)d\mathbf{W}_t. \tag{49}$$

Additionally, we need to show that the h-transformed transition density function satisfies the symmetric constraints. First, we show that if $h(\cdot, t_0)$ is SE(3)-invariant, then $h(\cdot, t)$ is also SE(3)-invariant $\forall t \in [0, T]$. For any $g \in$ SE(3), assume $\rho^{\mathcal{R}}(g)[z] = \mathbf{O}_{\mathcal{R}}(g)z + \mathbf{t}_{\mathcal{R}}$, where $\mathbf{O}_{\mathcal{R}}(g)$ is an orthogonal matrix and $\det(\mathbf{O}_{\mathcal{R}}(g)) = 1$. Since $h_{\mathcal{R}}(z, t)$ satisfies

$$h_{\mathcal{R}}(z, t) = \int p_{\mathcal{R}}(z', t_0|z, t)h(z', t_0)dz', \tag{50}$$

then we have

$$h_{\mathcal{R}}(\rho^{\mathcal{R}}(g)[z], t) = \int p_{\mathcal{R}}(z', t_0|\rho^{\mathcal{R}}(g)[z], t)h(z', t_0)dz' \tag{51}$$

$$= \int p_{\mathcal{R}}\left(\rho^{\mathcal{R}}(g)(\rho^{\mathcal{R}}(g))^{-1}[z'], t_0|\rho^{\mathcal{R}}(g)[z], t\right) h(\rho^{\mathcal{R}}(g)(\rho^{\mathcal{R}}(g))^{-1}[z'], t_0)dz'. \tag{52}$$

$$\tag{53}$$

By Proposition 3.1, $p_{\mathcal{R}}\left(\rho^{\mathcal{R}}(g)(\rho^{\mathcal{R}}(g))^{-1}[z'], t_0|\rho^{\mathcal{R}}(g)[z], t\right) = p_{\mathcal{R}}\left((\rho^{\mathcal{R}}(g))^{-1}[z'], t_0|z, t\right)$, let $z_1 = \rho^{\mathcal{R}}(g))^{-1}[z']$, then

$$h_{\mathcal{R}}(\rho^{\mathcal{R}}(g)[z], t) = \int p_{\mathcal{R}}\left((\rho^{\mathcal{R}}(g))^{-1}[z'], t_0|z, t\right) h(\rho^{\mathcal{R}}(g)(\rho^{\mathcal{R}}(g))^{-1}[z'], t_0)dz' \tag{54}$$

$$= \int p_{\mathcal{R}}(z_1, t_0|z, t) h(\rho^{\mathcal{R}}(g)z_1, t_0) \det(\mathbf{O}_{\mathcal{R}}(g))dz_1 \tag{55}$$

$$= \int p_{\mathcal{R}}(z_1, t_0|z, t) h(z_1, t_0)dz_1 \tag{56}$$

$$= h_{\mathcal{R}}(z, t). \tag{57}$$

So $h(\cdot, t)$ is SE(3)-invariant $\forall t \in [0, T]$, $h(\cdot, t)$ is well-defined under these symmetric constraints. Then we show $p_{\mathcal{R}}^h(z', t'|z, t)$ preserves the symmetric constraints:

$$p_{\mathcal{R}}^h(\rho^{\mathcal{R}}(g)[z'], t'|\rho^{\mathcal{R}}(g)[z], t) = p_{\mathcal{R}}(\rho^{\mathcal{R}}(g)[z'], t'|\rho^{\mathcal{R}}(g)[z], t)\frac{h_{\mathcal{R}}(\rho^{\mathcal{R}}(g)[z'], t')}{h_{\mathcal{R}}(\rho^{\mathcal{R}}(g)[z], t)} \tag{58}$$

$$= p_{\mathcal{R}}(z', t'|z, t)\frac{h_{\mathcal{R}}(\rho^{\mathcal{R}}(g)[z'], t')}{h_{\mathcal{R}}(\rho^{\mathcal{R}}(g)[z], t)} \tag{59}$$

$$= p_{\mathcal{R}}(z', t'|z, t)\frac{h_{\mathcal{R}}(z', t')}{h_{\mathcal{R}}(z, t)} \tag{60}$$

$$= p_{\mathcal{R}}^h(z', t'|z, t). \tag{61}$$

Thus we have proved that

$$p_{\mathcal{R}}^h(\rho^{\mathcal{R}}(g)[z'], t'|\rho^{\mathcal{R}}(g)[z], t) = p_{\mathcal{R}}^h(z', t'|z, t), \tag{62}$$

which implies that $p_{\mathcal{R}}^h(z', t'|z, t)$ preserves the symmetric constraints for any $g \in$ SE(3). So the proof is completed. $\qquad\square$

Next, we show how to construct a SDE with a fixed terminal point as an simple application of the Doob's h-transform. The result of this example is very useful to construct diffusion bridge.

**Proposition B.7.** *Assume the original SDE is given by* $d\mathbf{X}^t = \mathbf{f}(\mathbf{X}^t, t)dt + \sigma(t)d\mathbf{W}_t$. *Let* $h_{\mathcal{R}}(x, t) = p_{\mathcal{R}}(y, T|x, t)$ *which is the transition density function of the original SDE evaluated at* $\mathbf{X}_T = y$. *Then the h-transformed SDE*

$$d\mathbf{R}^t = \left[\mathbf{f}(\mathbf{R}^t, t) + \sigma^2(t)\nabla_{\mathbf{R}^t} \log p_{\mathcal{R}}(y, T|\mathbf{R}^t, t)\right] dt + \sigma(t)d\mathbf{W}_t, \tag{63}$$

*arrive at* $y$ *almost surely at the final time.*

*Proof.* The original SDE is given by

$$d\mathbf{X}^t = \mathbf{f}(\mathbf{X}^t, t)dt + \sigma(t)d\mathbf{W}_t. \tag{64}$$

First, we need to verify that $h_{\mathcal{R}}(x, t)$ satisfies the condition

$$h_{\mathcal{R}}(x, t) = \int p_{\mathcal{R}}(x', t_0 | x, t) h(x', t_0) dx'. \tag{65}$$

Since $h_{\mathcal{R}}(x, t) = p_{\mathcal{R}}(y, T | x, t)$, we have

$$\int p_{\mathcal{R}}(x', t' | x, t) h_{\mathcal{R}}(x', t') = \int p_{\mathcal{R}}(x', t' | x, t) p_{\mathcal{R}}(y, T | x', t') dx'. \tag{66}$$

Then by the Chapman–Kolmogorov's equation

$$\int p_{\mathcal{R}}(x', t' | x, t) p_{\mathcal{R}}(y, T | x', t') dx' = p_{\mathcal{R}}(y, T | x, t), \tag{67}$$

we get

$$\int p_{\mathcal{R}}(x', t' | x, t) h_{\mathcal{R}}(x', t') = p_{\mathcal{R}}(y, T | x, t) = h_{\mathcal{R}}(x, t). \tag{68}$$

So the condition is satisfied. Then we can use the result of the Proposition 3.2. The $h$-transformed SDE is given by

$$d\mathbf{R}^t = \left[ \mathbf{f}(\mathbf{R}^t, t) + \sigma^2(t) \nabla_{\mathbf{R}^t} \log p_{\mathcal{R}}(y, T | \mathbf{R}^t, t) \right] dt + \sigma(t) d\mathbf{W}_t. \tag{69}$$

And the h-transformed transition density function satisfies

$$\int_A p_{\mathcal{R}}^h(x', t' | x, t) dx' = \int_A p_{\mathcal{R}}(x', t' | x, t) \frac{h_{\mathcal{R}}(x', t')}{h_{\mathcal{R}}(x, t)} dx' \tag{70}$$

$$= \int_A p_{\mathcal{R}}(x', t' | x, t) \frac{p_{\mathcal{R}}(y, T | x', t')}{p_{\mathcal{R}}(y, T | x, t)} dx' \tag{71}$$

$$= P(\mathbf{X}_{t'} \in A | \mathbf{X}_t = x, \mathbf{X}_T = y), \tag{72}$$

where we use the Bayes' theorem to deduce the last equality and $A$ is an arbitrary Borel set. Since $\mathbf{R}_t$ is a process conditioning on $\mathbf{X}_T = y$, then $\mathbf{R}_T = y$ almost surly. $\square$

## B.4   Proof of Theorem 3.3

**Theorem B.8** (Equivariant Diffusion Bridge). *Given an SDE on geometric states* $d\mathbf{R}^t = \mathbf{f}_{\mathcal{R}}(\mathbf{R}^t, t)dt + \sigma(t)d\mathbf{W}^t$ *with transition density* $p_{\mathcal{R}}(z', t' | z, t), z, z' \in \mathcal{R}$ *satisfying the conditions in Proposition 3.1. Let* $h_{\mathcal{R}}(z, t; z_0) = \int p_{\mathcal{R}}(z', T | z, t) \frac{q_{data}(z' | z_0)}{p_{\mathcal{R}}(z', T | z_0, 0)} dz'$. *By using Proposition 3.2, we can derive the following* $h_{\mathcal{R}}$-*transformed SDE:*

$$d\mathbf{R}^t = \left[ \mathbf{f}_{\mathcal{R}}(\mathbf{R}^t, t) + \sigma^2(t) \mathbb{E}_{q_{\mathcal{R}}(\mathbf{R}^T, T | \mathbf{R}^t, t; \mathbf{R}^0)} [\nabla_{\mathbf{R}^t} \log p_{\mathcal{R}}(\mathbf{R}^T, T | \mathbf{R}^t, t) | \mathbf{R}^0, \mathbf{R}^t] \right] dt + \sigma(t) d\mathbf{W}^t, \tag{73}$$

*which corresponds to a process* $(\mathbf{R}^t)_{t \in [0, T]}, \mathbf{R}^0 \sim q_{data}(R^{t_0})$ *satisfying the following properties:*

- *let* $q(\cdot, \cdot) : \mathcal{R} \times \mathcal{R} \to \mathbb{R}_{\geq 0}$ *denote the joint distribution induced by* $(\mathbf{R}^t)_{t \in [0, T]}$, *then* $q(\mathbf{R}^0, \mathbf{R}^T)$ *equals to* $q_{data}(R^{t_0}, R^{t_1})$;

- *its transition density* $q_{\mathcal{R}}(\mathbf{R}^{t'}, t' | \mathbf{R}^t, t; \mathbf{R}^0) = q_{\mathcal{R}}(\rho^{\mathcal{R}}(g)[\mathbf{R}^{t'}], t' | \rho^{\mathcal{R}}(g)[\mathbf{R}^t], t; \rho^{\mathcal{R}}(g)[\mathbf{R}^0])$, $\forall 0 \leq t < t' \leq T, g \in \mathrm{SE}(3), \mathbf{R}^0 \sim q_{data}(R^{t_0})$.

*We call the tailored diffusion process* $(\mathbf{R}^t)_{t \in [0, T]}$ *an equivariant diffusion bridge.*

*Proof.* Let $h_{\mathcal{R}}(z, T; z_0) = \frac{q_{data}(z | z_0)}{p_{\mathcal{R}}(z, T | z_0, 0)}$, then we define

$$h_{\mathcal{R}}(z, t; z_0) = \int p_{\mathcal{R}}(z', T | z, t) \frac{q_{data}(z' | z_0)}{p_{\mathcal{R}}(z', T | z_0, 0)} dz', \forall t \in [0, T). \tag{74}$$

So we can easily show that $h_{\mathcal{R}}(z, t; z_0)$ satisfies the condition

$$h_{\mathcal{R}}(z, t; z_0) = \int p_{\mathcal{R}}(z', T|z, t) h(z', T; z_0) \mathrm{d}z', \forall t \in [0, T], \forall z, z_0 \in \mathcal{R}. \tag{75}$$

Then we can use the result of Theorem 3.2 to get the h-transformed SDE. By Theorem 3.2, the h-transformed SDE is

$$\mathrm{d}\mathbf{R}^t = \left[\mathbf{f}_{\mathcal{R}}(\mathbf{R}^t, t) + \sigma^2(t)\nabla_{\mathbf{R}^t} \log h_{\mathcal{R}}(\mathbf{R}^t, t; \mathbf{R}^0)\right] \mathrm{d}t + \sigma(t)\mathrm{d}\mathbf{W}_t. \tag{76}$$

Next, we need to find the explicit form of $\nabla_{\mathbf{R}^t} \log h_{\mathcal{R}}(\mathbf{R}^t, t; \mathbf{R}^0)$,

$$\nabla_z \log h_{\mathcal{R}}(z, t; z_0) = \frac{\nabla_z h_{\mathcal{R}}(z, t; z_0)}{h_z(z, t; z_0)} \tag{77}$$

$$= \frac{1}{h_{\mathcal{R}}(z, t; z_0)} \int \nabla_z p_{\mathcal{R}}(z', T|z, t) \frac{q_{\mathrm{data}}(z'|z_0)}{p_{\mathcal{R}}(z', T|z_0, 0)} \mathrm{d}z'. \tag{78}$$

The h-transformed density function is

$$q_{\mathcal{R}}(z', T|z, t; z_0, 0) = p_{\mathcal{R}}(z', T|z, t) \frac{h_{\mathcal{R}}(z', T; z_0)}{h_{\mathcal{R}}(z, t; z_0)} \tag{79}$$

$$= p_{\mathcal{R}}(z', T|z, t) \frac{q_{\mathrm{data}}(z'|z_0)}{p_{\mathcal{R}}(z', T; z_0, 0) h_{\mathcal{R}}(z, t; z_0)}. \tag{80}$$

Then we have

$$\nabla_z \log h_{\mathcal{R}}(z, t; z_0) = \frac{1}{h_{\mathcal{R}}(z, t; z_0)} \int \nabla_z p_{\mathcal{R}}(z', T|z, t) \frac{q_{\mathrm{data}}(z'|z_0)}{p_{\mathcal{R}}(z', T|z_0, 0)} \mathrm{d}z' \tag{81}$$

$$= \int \nabla_z p_{\mathcal{R}}(z', T|z, t) \frac{q_{\mathcal{R}}(z', T|z, t; z_0, 0)}{p_{\mathcal{R}}(z', T|z, t)} \mathrm{d}z' \tag{82}$$

$$= \int \nabla_z \log p_{\mathcal{R}}(z', T|z, t) q_{\mathcal{R}}(z', T|z, t; z_0, 0) \mathrm{d}z'. \tag{83}$$

So we get a explicit form of $\nabla_{\mathbf{R}^t} \log h_{\mathcal{R}}(\mathbf{R}^t, t; \mathbf{R}^0)$:

$$\nabla_{\mathbf{R}^t} \log h_{\mathcal{R}}(\mathbf{R}^t, t; \mathbf{R}^0) = \mathbb{E}_{q_{\mathcal{R}}(\mathbf{R}^T, T|\mathbf{R}^t, t; z_0)}[\nabla_{\mathbf{R}^t} \log p_{\mathcal{R}}(\mathbf{R}^T, T|\mathbf{R}^t, t)|z_0, \mathbf{R}^t]. \tag{84}$$

Then the h-transformed SDE becomes

$$\mathrm{d}\mathbf{R}^t = \left[\mathbf{f}_{\mathcal{R}}(\mathbf{R}^t, t) + \sigma^2(t)\mathbb{E}_{q_{\mathcal{R}}(\mathbf{R}^T, T|\mathbf{R}^t, t; z_0)}[\nabla_{\mathbf{R}^t} \log p_{\mathcal{R}}(\mathbf{R}^T, T|\mathbf{R}^t, t)|z_0, \mathbf{R}^t]\right] \mathrm{d}t + \sigma(t)\mathrm{d}\mathbf{W}^t. \tag{85}$$

Since $h_{\mathcal{R}}(z, 0; z_0) = \int p_{\mathcal{R}}(z', T|z, 0) \frac{q_{\mathrm{data}}(z'|z_0)}{p_{\mathcal{R}}(z', T|z_0, 0)} \mathrm{d}z' = \int q_{\mathrm{data}}(z'|z_0)\mathrm{d}z' = 1$, then

$$q_{\mathcal{R}}(z', T|z_0, 0) = p_{\mathcal{R}}(z', T|z_0, 0) \frac{q_{\mathrm{data}}(z'|z_0)}{p_{\mathcal{R}}(z', T; z_0, 0) h_{\mathcal{R}}(z_0, 0; z_0)} = q_{\mathrm{data}}(z'|z_0), \tag{86}$$

which means $q_{\mathcal{R}}(\mathbf{R}^T, T|\mathbf{R}^0, 0) = q_{\mathrm{data}}(\mathbf{R}^T|\mathbf{R}^0)$. Since the initial distribution $\mathbf{R}^0 \sim q_{\mathrm{data}}(R^{t_0})$, so $q_{\mathcal{R}}(\mathbf{R}^0) = q_{\mathrm{data}}(\mathbf{R}^0)$. So we can deduce that

$$q(\mathbf{R}^0, \mathbf{R}^T) = q_{\mathcal{R}}(\mathbf{R}^0) q_{\mathcal{R}}(\mathbf{R}^T, T|\mathbf{R}^0, 0) = q_{\mathrm{data}}(\mathbf{R}^0) q_{\mathrm{data}}(\mathbf{R}^T|\mathbf{R}^0) = q_{\mathrm{data}}(\mathbf{R}^0, \mathbf{R}^T). \tag{87}$$

Finally, we need to show that the transition density function satisfies the corresponding symmetric constrains. Since $h_{\mathcal{R}}(z', T; z_0) = \frac{q_{\mathrm{data}}(z'|z_0)}{p_{\mathcal{R}}(z', T|z_0, 0)}$ is SE(3)-invariant, i.e.

$$h_{\mathcal{R}}(\rho^{\mathcal{R}}(g)[z], T; \rho^{\mathcal{R}}(g)[z_0]) = h_{\mathcal{R}}(z', T; z_0), \forall g \in \mathrm{SE}(3), \tag{88}$$

we can show that $h(\cdot, t; \cdot)$ is also SE(3)-invariant $\forall t \in [0, T]$ using the following property

$$h_{\mathcal{R}}(z, t; z_0) = \int p_{\mathcal{R}}(z', T|z, t) h(z', T; z_0) \mathrm{d}z'. \tag{89}$$

For any $g \in \mathrm{SE}(3)$, assume $\rho^{\mathcal{R}}(g)[z] = \mathbf{O}_{\mathcal{R}}(g)z + \mathbf{t}_{\mathcal{R}}$, where $\mathbf{O}_{\mathcal{R}}(g)$ is an orthogonal matrix satisfying $\det(\mathbf{O}_{\mathcal{R}}(g)) = 1$, then we have

$$h_{\mathcal{R}}(\rho^{\mathcal{R}}(g)[z], t; \rho^{\mathcal{R}}(g)[z_0]) = \int p_{\mathcal{R}}(z', T | \rho^{\mathcal{R}}(g)[z], t) h(z', T; \rho^{\mathcal{R}}(g)[z_0]) \mathrm{d}z' \tag{90}$$

$$= \int p_{\mathcal{R}}\left(\rho^{\mathcal{R}}(g)(\rho^{\mathcal{R}}(g))^{-1}[z'], T | \rho^{\mathcal{R}}(g)[z], t\right) h(\rho^{\mathcal{R}}(g)(\rho^{\mathcal{R}}(g))^{-1}[z'], T; \rho^{\mathcal{R}}(g)[z_0]) \mathrm{d}z'. \tag{91}$$

$$\tag{92}$$

By Proposition 3.1, $p_{\mathcal{R}}\left(\rho^{\mathcal{R}}(g)(\rho^{\mathcal{R}}(g))^{-1}[z'], t_0 | \rho^{\mathcal{R}}(g)[z], t\right) = p_{\mathcal{R}}\left((\rho^{\mathcal{R}}(g))^{-1}[z'], t_0 | z, t\right)$, let $z_1 = \rho^{\mathcal{R}}(g))^{-1}[z']$, then

$$h_{\mathcal{R}}(\rho^{\mathcal{R}}(g)[z], t; \rho^{\mathcal{R}}(g)[z_0]) \tag{93}$$

$$= \int p_{\mathcal{R}}\left((\rho^{\mathcal{R}}(g))^{-1}[z'], T | z, t\right) h(\rho^{\mathcal{R}}(g)(\rho^{\mathcal{R}}(g))^{-1}[z'], T; \rho^{\mathcal{R}}(g)[z_0]) \mathrm{d}z' \tag{94}$$

$$= \int p_{\mathcal{R}}(z_1, T | z, t) h(\rho^{\mathcal{R}}(g)[z_1], T; \rho^{\mathcal{R}}(g)[z_0]) \det(\mathbf{O}_{\mathcal{R}}(g)) \mathrm{d}z_1 \tag{95}$$

$$= \int p_{\mathcal{R}}(z_1, t_0 | z, t) h(z_1, t_0; z_0) \mathrm{d}z_1 \tag{96}$$

$$= h_{\mathcal{R}}(z, t; z_0). \tag{97}$$

So $h(\cdot, t; \cdot)$ is $\mathrm{SE}(3)$-invariant $\forall t \in [0, T]$. Then we show $q_{\mathcal{R}}(z', t' | z, t; z_0, 0)$ preserves the symmetric constraints:

$$q_{\mathcal{R}}(\rho^{\mathcal{R}}(g)[z'], t' | \rho^{\mathcal{R}}(g)[z], t; \rho^{\mathcal{R}}(g)[z_0], 0) \tag{98}$$

$$= p_{\mathcal{R}}(\rho^{\mathcal{R}}(g)[z'], t' | \rho^{\mathcal{R}}(g)[z], t) \frac{h_{\mathcal{R}}(\rho^{\mathcal{R}}(g)[z'], t'; \rho^{\mathcal{R}}(g)[z_0])}{h_{\mathcal{R}}(\rho^{\mathcal{R}}(g)[z], t; \rho^{\mathcal{R}}(g)[z_0])} \tag{99}$$

$$= p_{\mathcal{R}}(z', t' | z, t) \frac{h_{\mathcal{R}}(\rho^{\mathcal{R}}(g)[z'], t'; \rho^{\mathcal{R}}(g)[z_0])}{h_{\mathcal{R}}(\rho^{\mathcal{R}}(g)[z], t; \rho^{\mathcal{R}}(g)[z_0])} \tag{100}$$

$$= p_{\mathcal{R}}(z', t' | z, t) \frac{h_{\mathcal{R}}(z', t'; z_0)}{h_{\mathcal{R}}(z, t; z_0)} \tag{101}$$

$$= q_{\mathcal{R}}(z', t' | z, t; z_0, 0), \tag{102}$$

which completes our proof. $\square$

## B.5 Objective Function of the Equivariant Diffusion Bridge

**Lemma B.9.** *Let* $\mathbf{X}_1, \cdots, \mathbf{X}_n, \mathbf{Y}, \mathbf{Z}$ *be random variables. Then the optimal approximation of* $\mathbf{Y}$ *based on* $\{\mathbf{X}\}_{i=1}^n$ *is* $f^*(\mathbf{X}_1, \cdots, \mathbf{X}_n) = \arg\min_f \mathbb{E}\|\mathbf{Y} - f(\mathbf{X}_1, \cdots, \mathbf{X}_n)\|^2 = \mathbb{E}[\mathbf{Y} | \mathbf{X}_1, \cdots, \mathbf{X}_n].$

*Proof.* Denote $\mathbf{X} = (\mathbf{X}_1, \cdots, \mathbf{X}_n)$. We show the following decomposition first:

$$\mathbb{E}\|\mathbf{Y} - f(\mathbf{X})\|^2 = \mathbb{E}\|\mathbf{Y} - \mathbb{E}[\mathbf{Y} | \mathbf{X}]\|^2 + \mathbb{E}\left[\|\mathbb{E}[\mathbf{Y} | \mathbf{X}] - f(\mathbf{X})\|^2\right]. \tag{103}$$

We can compute $\mathbb{E}\|\mathbf{Y} - f(\mathbf{X})\|^2$ directly by

$$\mathbb{E}\|\mathbf{Y} - f(\mathbf{X})\|^2 = \mathbb{E}\|\mathbf{Y} - \mathbb{E}[\mathbf{Y} | \mathbf{X}] + \mathbb{E}[\mathbf{Y} | \mathbf{X}] - f(\mathbf{X})\|^2 \tag{104}$$

$$= \mathbb{E}\|\mathbf{Y} - \mathbb{E}[\mathbf{Y} | \mathbf{X}]\|^2 + \mathbb{E}\left[\|\mathbb{E}[\mathbf{Y} | \mathbf{X}] - f(\mathbf{X})\|^2\right] \tag{105}$$

$$+ \mathbb{E}\langle \mathbf{Y} - \mathbb{E}[\mathbf{Y} | \mathbf{X}], \mathbb{E}[\mathbf{Y} | \mathbf{X}] - f(\mathbf{X})\rangle. \tag{106}$$

Since

$$\mathbb{E}\langle \mathbf{Y} - \mathbb{E}[\mathbf{Y} | \mathbf{X}], \mathbb{E}[\mathbf{Y} | \mathbf{X}] - f(\mathbf{X})\rangle = \mathbb{E}\left[\mathbb{E}\langle \mathbf{Y} - \mathbb{E}[\mathbf{Y} | \mathbf{X}], \mathbb{E}[\mathbf{Y} | \mathbf{X}] - f(\mathbf{X})\rangle | \mathbf{X}\right] = 0, \tag{107}$$

we have

$$\mathbb{E}\|\mathbf{Y} - f(\mathbf{X})\|^2 = \mathbb{E}\|\mathbf{Y} - \mathbb{E}[\mathbf{Y} | \mathbf{X}]\|^2 + \mathbb{E}\left[\|\mathbb{E}[\mathbf{Y} | \mathbf{X}] - f(\mathbf{X})\|^2\right] \tag{108}$$

$$+ \mathbb{E}\langle \mathbf{Y} - \mathbb{E}[\mathbf{Y} | \mathbf{X}], \mathbb{E}[\mathbf{Y} | \mathbf{X}] - f(\mathbf{X})\rangle \tag{109}$$

$$= \mathbb{E}\|\mathbf{Y} - \mathbb{E}[\mathbf{Y} | \mathbf{X}]\|^2 + \mathbb{E}\left[\|\mathbb{E}[\mathbf{Y} | \mathbf{X}] - f(\mathbf{X})\|^2\right] \tag{110}$$

$$\geq \mathbb{E}\|\mathbf{Y} - \mathbb{E}[\mathbf{Y} | \mathbf{X}_1, \cdots, \mathbf{X}_n]\|^2. \tag{111}$$

The inequality becomes equality if and only if $f(\mathbf{X}_1, \cdots, \mathbf{X}_n) = \mathbb{E}[\mathbf{Y}|\mathbf{X}_1, \cdots, \mathbf{X}_n]$. So the the optimal approximation of $\mathbf{Y}$ based on $\{\mathbf{X}\}_{i=1}^n$ is $\mathbb{E}[\mathbf{Y}|\mathbf{X}_1, \cdots, \mathbf{X}_n]$, i.e.

$$f^*(\mathbf{X}_1, \cdots, \mathbf{X}_n) = \arg\min_f \mathbb{E}\|\mathbf{Y} - f(\mathbf{X}_1, \cdots, \mathbf{X}_n)\|^2 = \mathbb{E}[\mathbf{Y}|\mathbf{X}_1, \cdots, \mathbf{X}_n]. \tag{112}$$

$\square$

**Proposition B.10.** *The training objective function of Equivariant Diffusion Bridge is:*

$$\mathcal{L}(\theta) = \mathbb{E}_{(z_0,z_1)\sim q_{data}(R^{t_0}, R^{t_1}), \mathbf{R}^t \sim q_{\mathcal{R}}(\mathbf{R}^t, t|z_1, T; z_0, 0)} \lambda(t) \|\mathbf{v}_\theta(\mathbf{R}^t, t; z_0) - \nabla_{\mathbf{R}^t} \log p_{\mathcal{R}}(z_1, T|\mathbf{R}^t, t)\|^2, \tag{113}$$

*where $t \sim \mathcal{U}(0,T)$. Then the optimal parameter $\theta^* = \arg\min_\theta \mathcal{L}(\theta)$ satisfies*

$$\mathbf{v}_{\theta^*}(\mathbf{R}^t, t; z_0) = \mathbb{E}_{q_{\mathcal{R}}(\mathbf{R}^T, T|\mathbf{R}^t, t; \mathbf{R}^0)}[\nabla_{\mathbf{R}^t} \log p_{\mathcal{R}}(\mathbf{R}^T, T|\mathbf{R}^t, t)|\mathbf{R}^0, \mathbf{R}^t]. \tag{114}$$

*Proof.* Let $\mathcal{L}(\theta) = \mathbb{E}_{t\sim\mathcal{U}(0,T)}\lambda(t)\mathcal{L}^t(\theta)$, where

$$\mathcal{L}^t(\theta) = \mathbb{E}_{(z_0,z_1)\sim q_{data}(R^{t_0}, R^{t_1}), \mathbf{R}^t \sim q_{\mathcal{R}}(\mathbf{R}^t, t|z_1, T; z_0, 0)} \|\mathbf{v}_\theta(\mathbf{R}^t, t; z_0) - \nabla_{\mathbf{R}^t} \log p_{\mathcal{R}}(z_1, T|\mathbf{R}^t, t)\|^2. \tag{115}$$

Then by Lemma B.9, $\mathbf{v}_\theta(\mathbf{R}^t, t; z_0) = \mathbb{E}_{q_{\mathcal{R}}(\mathbf{R}^T, T|\mathbf{R}^t, t; \mathbf{R}^0)}[\nabla_{\mathbf{R}^t} \log p_{\mathcal{R}}(\mathbf{R}^T, T|\mathbf{R}^t, t)|\mathbf{R}^0, \mathbf{R}^t]$ minimize $\mathcal{L}^t(\theta), \forall t \in [0,T]$. Since $\lambda(t) \geq 0$, so the optimal parameter $\theta^* = \arg\min_\theta \mathcal{L}(\theta)$ satisfies

$$\mathbf{v}_{\theta^*}(\mathbf{R}^t, t; z_0) = \mathbb{E}_{q_{\mathcal{R}}(\mathbf{R}^T, T|\mathbf{R}^t, t; \mathbf{R}^0)}[\nabla_{\mathbf{R}^t} \log p_{\mathcal{R}}(\mathbf{R}^T, T|\mathbf{R}^t, t)|\mathbf{R}^0, \mathbf{R}^t], \forall t \in [0,T]. \tag{116}$$

$\square$

## B.6 Proof of Theorem 3.4

**Theorem B.11** (Chain of Equivariant Diffusion Bridges). *Let $\{(\mathbf{R}_i^t)_{t\in[0,T]}\}_{i\in[N-1]}$ denote a series of $N$ equivaraint diffusion bridges defined in Theorem 3.3. For the $i$-th bridge $(\mathbf{R}_i^t)_{t\in[0,T]}$, if we set (1) $h_{\mathcal{R}}^i(z, t; z_0) = \int p_{\mathcal{R}}(z', T|z, t) \frac{q_{traj}^{i+1}(z'|z_0)}{p_{\mathcal{R}}(z', T|z_0, 0)} dz'$; (2) $\mathbf{R}_0^0 \sim q_{traj}^0(\tilde{R}^0), \mathbf{R}_i^0 = \mathbf{R}_{i-1}^T, \forall 0 < i < N$, then the joint distribution $q_{\mathcal{R}}(\mathbf{R}_0^0, \mathbf{R}_0^T, \mathbf{R}_1^T, \cdots, \mathbf{R}_{N-1}^T)$ induced by $\{(\mathbf{R}^t)_{t\in[0,T]}\}_{i\in[N-1]}$ equals to $q_{traj}(\tilde{R}^0, ..., \tilde{R}^N)$. We call this process a chain of equivariant diffusion bridges.*

*Proof.* By Theorem 3.3, the transition density function of $(\mathbf{R}_i^t)_{t\in[0,T]}$ satisfies $q_{\mathcal{R}}^i(\mathbf{R}_i^T|\mathbf{R}_i^0) = q_{traj}^i(\mathbf{R}_i^T|\mathbf{R}_i^0), \forall 0 \leq i \leq N-1$. The ground truth probability density function has the decomposition $q_{traj}^0(\tilde{R}^0)\prod_{i=1}^N q_{traj}^i(\tilde{R}^i|\tilde{R}^{i-1})$. Then we use the boundary condition, $\mathbf{R}_0^0 \sim q_{traj}^0(\tilde{R}^0), \mathbf{R}_i^0 = \mathbf{R}_{i-1}^T, \forall 0 < i < N$, we have

$$q(\mathbf{R}_0^0, \mathbf{R}_0^T, \mathbf{R}_1^T, \cdots, \mathbf{R}_{N-1}^T) = q_{\mathcal{R}}^0(\mathbf{R}_0^0)\prod_{i=1}^N q_{\mathcal{R}}^i(\mathbf{R}_i^T|\mathbf{R}_{i-1}^T) \tag{117}$$

$$= q_{\mathcal{R}}^0(\mathbf{R}_0^0)\prod_{i=1}^N q_{\mathcal{R}}^i(\mathbf{R}_i^T|\mathbf{R}_i^0) \tag{118}$$

$$= q_{traj}^0(\mathbf{R}_0^0)\prod_{i=1}^N q_{traj}^i(\mathbf{R}_i^T|\mathbf{R}_i^0) \tag{119}$$

$$= q_{traj}(\mathbf{R}_0^0, \mathbf{R}_0^T, \mathbf{R}_1^T, \cdots, \mathbf{R}_{N-1}^T). \tag{120}$$

So the joint distribution $q_{\mathcal{R}}(\mathbf{R}_0^0, \mathbf{R}_0^T, \mathbf{R}_1^T, \cdots, \mathbf{R}_{N-1}^T)$ induced by $\{(\mathbf{R}^t)_{t\in[0,T]}\}_{i\in[N-1]}$ equals to $q_{traj}(\tilde{R}^0, ..., \tilde{R}^N)$. $\square$

## B.7 Objective of the Chain of Equivariant Diffusion Bridge

**Proposition B.12.** *The training objective function of the Chain of Equivariant Diffusion Bridge is:*

$$\mathcal{L}'(\theta) = \mathbb{E}_{(z_0,\dots,z_N)\sim q_{traj}(\tilde{R}^0,\dots,\tilde{R}^N),t,\mathbf{R}_i^{t'}} \lambda(t)\|\mathbf{v}_\theta(\mathbf{R}_i^{t'},t;z_i) - \nabla_{\mathbf{R}_i^{t'}} \log p_{\mathcal{R}}^i(z_{i+1},T|\mathbf{R}_i^{t'},t')\|^2, \tag{121}$$

*where $t \sim \mathcal{U}(0, N \times T), i = \lfloor \frac{t}{T} \rfloor, t' = t - i \times T, \mathbf{R}_i^{t'} \sim q_{\mathcal{R}}^i(\mathbf{R}^{t'},t'|z_{i+1},T;z_i,0)$. Then the optimal parameter $\theta^* = \arg\min_\theta \mathcal{L}'(\theta)$ satisfies*

$$\mathbf{v}_{\theta^*}(\mathbf{R}_i^{t'},t;z_i) = \mathbb{E}_{q_{\mathcal{R}}^i(\mathbf{R}_i^T,T|\mathbf{R}_i^{t'},t;\mathbf{R}_i^0)}[\nabla_{\mathbf{R}_i^t} \log p_{\mathcal{R}}^i(\mathbf{R}_i^T,T|\mathbf{R}_i^t,t)|\mathbf{R}_i^0,\mathbf{R}_i^t]. \tag{122}$$

*Proof.* Let $\mathcal{L}'(\theta) = \mathbb{E}_{t\sim\mathcal{U}(0,NT)}\lambda(t)\mathcal{L}_t'(\theta)$, where

$$\mathcal{L}_t'(\theta) = \mathbb{E}_{(z_0,\dots,z_N)\sim q_{traj}(\tilde{R}^0,\dots,\tilde{R}^N),t,\mathbf{R}_i^{t'}} \|\mathbf{v}_\theta(\mathbf{R}_i^{t'},t;z_i) - \nabla_{\mathbf{R}_i^{t'}} \log p_{\mathcal{R}}^i(z_{i+1},T|\mathbf{R}_i^{t'},t')\|^2, \tag{123}$$

where $t \sim \mathcal{U}(0, N \times T), i = \lfloor \frac{t}{T} \rfloor, t' = t - i \times T, \mathbf{R}_i^{t'} \sim q_{\mathcal{R}}^i(\mathbf{R}^{t'},t'|z_{i+1},T;z_i,0)$. Then by Lemma B.9, $\mathbf{v}_{\theta^*}(\mathbf{R}_i^{t'},t;z_i) = \mathbb{E}_{q_{\mathcal{R}}^i(\mathbf{R}_i^T,T|\mathbf{R}_i^{t'},t;\mathbf{R}_i^0)}[\nabla_{\mathbf{R}_i^t} \log p_{\mathcal{R}}^i(\mathbf{R}_i^T,T|\mathbf{R}_i^t,t)|\mathbf{R}_i^0,\mathbf{R}_i^t]$ minimize $\mathcal{L}_t'(\theta), \forall t \in [0, NT]$. Since $\lambda(t) \geq 0$, so the optimal parameter $\theta^* = \arg\min_\theta \mathcal{L}(\theta)$ satisfies

$$\mathbf{v}_{\theta^*}(\mathbf{R}_i^{t'},t;z_i) = \mathbb{E}_{q_{\mathcal{R}}^i(\mathbf{R}_i^T,T|\mathbf{R}_i^{t'},t;\mathbf{R}_i^0)}[\nabla_{\mathbf{R}_i^t} \log p_{\mathcal{R}}^i(\mathbf{R}_i^T,T|\mathbf{R}_i^t,t)|\mathbf{R}_i^0,\mathbf{R}_i^t]. \tag{124}$$

$\square$

## B.8 Proof of Theorem 3.5

In this paper, we choose the Brownian bridge as our matching target. Let's first recall the definition and properties of the Brownian bridge. A Brownian bridge $(X_t)_{t\in[0,T]}$ with the initial position $X_0$ and the terminal position $X_T$ is given by the following SDE

$$d\mathbf{X}_t = \frac{\mathbf{X}_T - \mathbf{X}_t}{T - t}dt + \sigma d\mathbf{W}_t, \tag{125}$$

where $\mathbf{W}_t$ is the standard wiener process. The solution of the Brownian bridge is given by

$$\mathbf{X}_t \sim \mathcal{N}\left((1-t)\mathbf{X}_0 + t\mathbf{X}_1, \sigma^2 t(1-t)\right). \tag{126}$$

Next, we recall the definition of the KL Divergence:

**Definition B.13** (KL Divergence). *The relative entropy (or Kullback–Leibler Divergence) $\mathrm{KL}(f\|g)$ between two probability density functions $f(x)$ and $g(x)$ is defined by:*

$$\mathrm{KL}(f\|g) = \int f(x) \log \frac{f(x)}{g(x)}dx. \tag{127}$$

*In general, let $\mathbb{P}$ and $\mathbb{Q}$ be two probability measures on space $\mathcal{X}$. Assume $\mathbb{P}$ is absolutely continuous with respect to $\mathbb{Q}$ then the Kullback–Leibler Divergence between $\mathbb{P}$ and $\mathbb{Q}$ is defined as follows*

$$\mathrm{KL}(\mathbb{P}\|\mathbb{Q}) = \int_{\mathcal{X}} \log \frac{d\mathbb{P}}{d\mathbb{Q}}d\mathbb{P}, \tag{128}$$

*where $\frac{d\mathbb{P}}{d\mathbb{Q}}$ is the Radon–Nikodym derivative of $\mathbb{P}$ with respect to $\mathbb{Q}$.*

When we need to compute the KL divergence between the path measures associated with two SDEs, the Girsanov's theorem [53] is an useful tool to get the Radon–Nikodym derivative between the two measure. The precise statements are as follows.

**Theorem B.14** (Girsanov's Theorem). *Let $\mathbf{W}_t$ be a $d$-dimensional Wiener process defined on $(\Omega, \mathcal{F}, (\mathcal{F}_t), \mathbb{P})$. Let $\mathbf{H}_t$ be a $d$-dimensional $\mathcal{F}_t-$adapted process such that*

$$\int_0^T \|\mathbf{H}_t\|^2 dt < \infty, \mathbb{P} - a.s. \tag{129}$$

*Define*

$$Z_t = \exp\left(\int_0^t \mathbf{H}_s \cdot \mathrm{d}\mathbf{W}_t - \frac{1}{2}\int_0^t \|\mathbf{H}_t\|^2 \mathrm{d}s\right). \tag{130}$$

*Assume $Z_t$ is a martingale. Define the probability measure $\mathbb{Q}$ on $\mathcal{F}_T$ by*

$$\mathrm{d}\mathbb{Q} = Z_T \mathrm{d}\mathbb{P}. \tag{131}$$

*Let $\mathbf{M}_t = \mathbf{W}_t - \int_0^t \mathbf{H}_s \mathrm{d}s$, then $\mathbf{M}_t$ is a $d-$dimensional Wiener process with respect to $\mathbb{Q}$.*

In practice, the condition that $Z_t$ is a martingale is hard to vertify. So the condition is often replaced by the Novikov's condition

$$\mathbb{E}\left[\exp\left(\frac{1}{2}\int_0^T \|\mathbf{H}_t\|^2 \mathrm{d}t\right)\right] < \infty. \tag{132}$$

For more discussions and applications of the Girsanov's theorem, please see [86, 74, 28]. Now we can give the precise assumptions and proof of Theorem 3.5 using the properties of Brownian Bridge and Theorem B.14.

**Theorem B.15.** *Assume $(\tilde{R}^i)_{i \in [N]}$ is sampled by simulating a prior SDE on geometric states $\mathrm{d}\tilde{\mathbf{R}}^t = -\nabla H_{\mathcal{R}}^*(\tilde{\mathbf{R}}^t)\mathrm{d}t + \sigma \mathrm{d}\tilde{\mathbf{W}}^t$. Let $\mu_i^*$ denote the path measure of this prior SDE when $t \in [iT, (i+1)T]$. Building upon $(\tilde{R}^i)_{i \in [N]}$, let $\{\mu_{\mathcal{R}}^i\}_{i \in [N-1]}$ denote the path measure of our chain of equivariant diffusion bridges. Assume $\{\mu_{\mathcal{R}}^i\}_{i \in [N-1]}$ is composed of chain of the Brownian Bridge. Assume the total time is $NT = 1$. Under the following assumptions:*

- *$H_{\mathcal{R}}^*(\cdot) : \mathbb{R}^d \to \mathbb{R}$ is a scalar function with continuous second-order partial derivative;*

- *The drift function is Lipschitz: there exist a constant $L$ such that*
$$\|\nabla H_{\mathcal{R}}^*(x) - \nabla H_{\mathcal{R}}^*(y)\| \le L\|x - y\|, \forall x, y \in \mathbb{R}^d;$$

- *$H_{\mathcal{R}}^*(\cdot)$ satisfies $\|\nabla H_{\mathcal{R}}^*(x)\| \le K(1 + \|x\|), \forall x \in \mathbb{R}^d$;*

- *$\mathbb{E}\|\tilde{\mathbf{R}}^t\|^2 < M, \forall t \in [0, NT]$;*

- *$h(t) = \mathbb{E}[H_{\mathcal{R}}^*(\tilde{\mathbf{R}}^t)]$ is a continuous function on $t \in [0, NT]$;*

- *The Novikov's condition:*
$$\mathbb{E}\left[\exp\left(\frac{1}{2}\int_0^{NT} \|\nabla H_{\mathcal{R}}^*(\tilde{\mathbf{W}}^t)\|^2 \mathrm{d}t\right)\right] < \infty;$$

- *The function $H_{\mathcal{R}}^*$ satisfies the following regulaity condition: there exist a constant $C$ such that $\nabla^2 H_{\mathcal{R}}^*(x) - \|\nabla H_{\mathcal{R}}^*(x)\|^2/\sigma^2 < C, \forall x \in \mathbb{R}^d$;*

*then we have $\lim\limits_{N \to \infty} \max\limits_i \mathrm{KL}(\mu_i^* \| \mu_{\mathcal{R}}^i) = 0$.*

*Proof.* Let $p^*$ be the probability density function associated with the ground truth SDE $\mathrm{d}\tilde{\mathbf{R}}^t = \mathbf{f}_{\mathcal{R}}^*(\tilde{\mathbf{R}}^t, t)\mathrm{d}t + \sigma \mathrm{d}\tilde{\mathbf{W}}^t, \tilde{\mathbf{R}}^0 = \mathbf{R}^0$. Let $\{(\mathbf{R}_i^t)_{t \in [0,T]}\}_{i \in [N-1]}$ denote a series of $N$ equivaraint diffusion bridges defined in Theorem 3.4. Then by theorem 3.4, $q_{\mathcal{R}}(\mathbf{R}_0^0, \mathbf{R}_0^T, \mathbf{R}_1^T, \cdots, \mathbf{R}_{N-1}^T)$ induced by $\{(\mathbf{R}^t)_{t \in [0,T]}\}_{i \in [N-1]}$ equals to $p_{\mathcal{R}}^*(\mathbf{R}_0^0, \mathbf{R}_0^T, \mathbf{R}_1^T, \cdots, \mathbf{R}_{N-1}^T)$. Additionally, the conditional probability density function $q_{\mathcal{R}}(\mathbf{R}_i^t | \mathbf{R}_i^T, \mathbf{R}_i^0)$, for $iT \le t < (i+1)T$, is associated with the Brownian bridge

$$\mathrm{d}\mathbf{R}_i^t = \frac{\mathbf{R}_i^T - \mathbf{R}_i^t}{T - t'}\mathrm{d}t' + \sigma \mathrm{d}\mathbf{W}_t, \tag{133}$$

where $t' = t - iT$. Then by the chain rule of KL divergence

$$\mathrm{KL}(\mu_i^* \| \mu_{\mathcal{R}}^i) = \mathrm{KL}(p_i^*(\tilde{\mathbf{R}}^{(i+1)T}, \tilde{\mathbf{R}}^{iT}) \| q_{\mathcal{R}}^i(\tilde{\mathbf{R}}^{(i+1)T}, \tilde{\mathbf{R}}^{iT}) + \tag{134}$$

$$\mathbb{E}_{p_i^*(\tilde{\mathbf{R}}^{(i+1)T}, \tilde{\mathbf{R}}^{iT})}\left[\mathrm{KL}(\mu_i^*(\cdot | \tilde{\mathbf{R}}^{(i+1)T}, \tilde{\mathbf{R}}^{iT}) \| \mu_{\mathcal{R}}^i(\cdot | \tilde{\mathbf{R}}^{(i+1)T}, \tilde{\mathbf{R}}^{iT}))\right]. \tag{135}$$

Since $p_i^*(\tilde{\mathbf{R}}^{(i+1)T}, \tilde{\mathbf{R}}^{iT}) = q_{\mathcal{R}}^i(\tilde{\mathbf{R}}^{(i+1)T}, \tilde{\mathbf{R}}^{iT})$, we have

$$\mathrm{KL}(\mu_i^*||\mu_{\mathcal{R}}^i) = \mathbb{E}_{p_i^*(\tilde{\mathbf{R}}^{(i+1)T}, \tilde{\mathbf{R}}^{iT})}\left[\mathrm{KL}(\mu_i^*(\cdot|\tilde{\mathbf{R}}^{(i+1)T}, \tilde{\mathbf{R}}^{iT})||\mu_{\mathcal{R}}^i(\cdot|\tilde{\mathbf{R}}^{(i+1)T}, \tilde{\mathbf{R}}^{iT}))\right]. \quad (136)$$

Since the prior SDE is time homogeneous, we can only consider the case $i = 0$ without loss of generality. Let $\upsilon$ be the path measure of the Brownian motion $\sigma\tilde{\mathbf{W}}^t$ on space $\mathcal{R}$. Since the condition of Theorem B.14 is satisfied, then we can use Theorem B.14 and get

$$\mathrm{d}\mu_0^*(\cdot|\tilde{\mathbf{R}}^0) = \exp\left(-\frac{1}{\sigma}\int_0^T \nabla H_{\mathcal{R}}^*(\sigma\tilde{\mathbf{W}}^t)\cdot\mathrm{d}\tilde{\mathbf{W}}^t - \frac{1}{2\sigma^2}\int_0^T \|\nabla H_{\mathcal{R}}^*(\sigma\tilde{\mathbf{W}}^t)\|^2\mathrm{d}t\right)\mathrm{d}\upsilon(\cdot|\tilde{\mathbf{R}}^0). \tag{137}$$

Then we can use the Ito's formula (Theorem B.1) to simplify the expression

$$\mathrm{d}\mu_0^*(\cdot|\tilde{\mathbf{R}}^0) = \exp\left(\tfrac{1}{\sigma^2}(H_{\mathcal{R}}^*(\sigma\tilde{\mathbf{W}}^0) - H_{\mathcal{R}}^*(\sigma\tilde{\mathbf{W}}^T)) + \tfrac{1}{2}\int_0^T(\nabla^2 H_{\mathcal{R}}^*(\sigma\tilde{\mathbf{W}}^t) - \tfrac{1}{\sigma^2}\|\nabla H_{\mathcal{R}}^*(\sigma\tilde{\mathbf{W}}^t)\|^2)\mathrm{d}t\right)\mathrm{d}\upsilon(\cdot|\tilde{\mathbf{R}}^0). \tag{138}$$

To simplify our notation, we denote

$$Z_T = \exp\left(\frac{1}{\sigma^2}(H_{\mathcal{R}}^*(\sigma\tilde{\mathbf{W}}^0) - H_{\mathcal{R}}^*(\sigma\tilde{\mathbf{W}}^T)) + \frac{1}{2}\int_0^T(\nabla^2 H_{\mathcal{R}}^*(\sigma\tilde{\mathbf{W}}^t) - \frac{1}{\sigma^2}\|\nabla H_{\mathcal{R}}^*(\sigma\tilde{\mathbf{W}}^t)\|^2)\mathrm{d}t\right). \tag{139}$$

Let $F, g$ be measurable functions on $C[0, T], \mathbb{R}^d$, respectively. Then by the disintegration of Wiener measure into pinned Wiener measures (path measure of the Brownian Bridge), we have

$$\mathbb{E}_{\mu_0^*(\cdot|\tilde{\mathbf{R}}^0)}[Fg(\sigma\tilde{\mathbf{W}}^T)] = \mathbb{E}_{\upsilon(\cdot|\tilde{\mathbf{R}}^0)}[Fg(\sigma\tilde{\mathbf{W}}^T)Z_T] = \int \mathbb{E}_{\upsilon(\cdot|\tilde{\mathbf{R}}^0, \tilde{\mathbf{R}}^T=x)}[FZ_T]g(x)p_T(x|\tilde{\mathbf{R}}^0)\mathrm{d}x, \tag{140}$$

where $p_T(x|\tilde{\mathbf{R}}^0)$ is the transition density function of $\sigma\tilde{\mathbf{W}}^t$. Let $F = 1$, we get

$$\int \mathbb{E}_{\upsilon(\cdot|\tilde{\mathbf{R}}^0, \tilde{\mathbf{R}}^T=x)}[Z_T]g(x)p_T(x|\tilde{\mathbf{R}}^0)\mathrm{d}x = \int g(x)p_0^*(x|\tilde{\mathbf{R}}^0)\mathrm{d}x. \tag{141}$$

So we have $\mathbb{E}_{\upsilon(\cdot|\tilde{\mathbf{R}}^0, \tilde{\mathbf{R}}^T=x)}[Z_T] = p_0^*(x|\tilde{\mathbf{R}}^0)/p_T(x|\tilde{\mathbf{R}}^0)$. Let $g = 1$, then we get

$$\int \mathbb{E}_{\mu_0^*(\cdot|\tilde{\mathbf{R}}^0, \tilde{\mathbf{R}}^T=x)}[F]p_0^*(x|\tilde{\mathbf{R}}^0)\mathrm{d}x = \int \mathbb{E}_{\upsilon(\cdot|\tilde{\mathbf{R}}^0, \tilde{\mathbf{R}}^T=x)}[FZ_T]p_T(x|\tilde{\mathbf{R}}^0)\mathrm{d}x. \tag{142}$$

So we can conclude that

$$\frac{\mathrm{d}\mu_0^*(\cdot|\tilde{\mathbf{R}}^0, \tilde{\mathbf{R}}^T)}{\mathrm{d}\upsilon(\cdot|\tilde{\mathbf{R}}^0, \tilde{\mathbf{R}}^T)} = \frac{p_T(\tilde{\mathbf{R}}^T|\tilde{\mathbf{R}}^0)}{p_0^*(\tilde{\mathbf{R}}^T|\tilde{\mathbf{R}}^0)} \cdot \frac{\exp\left(\frac{1}{\sigma^2}(H_{\mathcal{R}}^*(\tilde{\mathbf{R}}^0))\right)}{\exp\left(\frac{1}{\sigma^2}(H_{\mathcal{R}}^*(\tilde{\mathbf{R}}^T))\right)} \exp\left(\frac{1}{2}\int_0^T(\nabla^2 H_{\mathcal{R}}^*(\cdot) - \frac{1}{\sigma^2}\|\nabla H_{\mathcal{R}}^*(\cdot)\|^2)\mathrm{d}t\right). \tag{143}$$

Note that $\mu_{\mathcal{R}}^i(\cdot|\tilde{\mathbf{R}}^0, \tilde{\mathbf{R}}^T) = \upsilon(\cdot|\tilde{\mathbf{R}}^0, \tilde{\mathbf{R}}^T)$. Now we can calculate the KL divergence by

$$\mathrm{KL}(\mu_0^*||\mu_{\mathcal{R}}^0) = \mathbb{E}_{p_0^*(\tilde{\mathbf{R}}^T, \tilde{\mathbf{R}}^0)}\left[\mathrm{KL}(\mu_0^*(\cdot|\tilde{\mathbf{R}}^T, \tilde{\mathbf{R}}^0)||\mu_{\mathcal{R}}^0(\cdot|\tilde{\mathbf{R}}^T, \tilde{\mathbf{R}}^0))\right] \tag{144}$$

$$\leq \mathbb{E}_{p_0^*(\tilde{\mathbf{R}}^T, \tilde{\mathbf{R}}^0)}\left[\log\left(\frac{p_T(\tilde{\mathbf{R}}^T|\tilde{\mathbf{R}}^0)}{p_0^*(\tilde{\mathbf{R}}^T|\tilde{\mathbf{R}}^0)} \cdot \frac{\exp\left(\frac{1}{\sigma^2}(H_{\mathcal{R}}^*(\tilde{\mathbf{R}}^0))\right)}{\exp\left(\frac{1}{\sigma^2}(H_{\mathcal{R}}^*(\tilde{\mathbf{R}}^T))\right)}\right)\right] + \frac{CT}{2} \tag{145}$$

$$= \mathbb{E}_{p_0^*(\tilde{\mathbf{R}}^0, \tilde{\mathbf{R}}^t)}\left[\log\left(\frac{p_T(\tilde{\mathbf{R}}^T|\tilde{\mathbf{R}}^0)}{p_0^*(\tilde{\mathbf{R}}^T|\tilde{\mathbf{R}}^0)}\right)\right] + \mathbb{E}[\frac{1}{\sigma^2}H_{\mathcal{R}}^*(\tilde{\mathbf{R}}^0)] - \mathbb{E}[\frac{1}{\sigma^2}H_{\mathcal{R}}^*(\tilde{\mathbf{R}}^T)] + \frac{CT}{2} \tag{146}$$

$$= -\mathbb{E}_{p_0^*(\tilde{\mathbf{R}}^0)}\mathrm{KL}(p_0^*(\tilde{\mathbf{R}}^T|\tilde{\mathbf{R}}^0)||p_T(\tilde{\mathbf{R}}^T|\tilde{\mathbf{R}}^0)) + \frac{h(0) - h(T)}{\sigma^2} + \frac{CT}{2} \tag{147}$$

$$\leq \frac{h(0) - h(T)}{\sigma^2} + \frac{CT}{2}. \tag{148}$$

When $N \to \infty$, $T = \frac{1}{N} \to 0$. Since $h(t)$ is continuous by our assumption, then $\mathrm{KL}(\mu_0^*||\mu_{\mathcal{R}}^0) \to 0$. $\qquad\square$

## C Derivation of Practical Objective Function

In this subsection, we show the implementation details of our framework. We set $T = 1$ in all the experiments.

**Matching objective.** We design the SDE on geometric states in Proposition 3.1 to be:

$$d\mathbf{R}^t = \sigma d\mathbf{W}^t, \quad \textit{with transition density} \quad p_{\mathcal{R}}(z', t'|z, t) = \mathcal{N}(z_0, \sigma^2(t' - t)\mathbf{I}) \tag{149}$$

The explicit form of the objective is

$$\nabla_{\mathbf{R}^t} \log p_{\mathcal{R}}(z_1, 1|\mathbf{R}^t, t) = \nabla_{\mathbf{R}^t} \log \mathcal{N}(z_0, \sigma^2(1 - t)\mathbf{I}) = \frac{z_1 - \mathbf{R}^t}{\sigma^2(1 - t)} \tag{150}$$

Then the h-transformed SDE becomes

$$d\mathbf{R}^t = \frac{\mathbf{R}^1 - \mathbf{R}^t}{1 - t} dt + \sigma d\mathbf{W}^t, \tag{151}$$

which is known as the Brownian bridge. The corresponding h-transformed density is

$$q_{\mathcal{R}}(\mathbf{R}^t, t|z_1, 1; z_0, 0) = \mathcal{N}(tz_1 + (1 - t)z_0, \sigma^2 t(1 - t)\mathbf{I}). \tag{152}$$

In practice, we do not use $q_{\mathcal{R}}(\mathbf{R}^0, 0|z_1, 1; z_0, 0) = \delta(\mathbf{R}^0 - z_0)$ as our initial distribution. We use $q_{\mathcal{R}}(\mathbf{R}^0, 0|z_1, 1; z_0, 0) = \mathcal{N}(z_0, \sigma^2 \mathbf{I})$ instead. Since the solution of the Brownian bridge is given by

$$\mathbf{R}^t = (1 - t)\mathbf{R}^0 + t\mathbf{R}^1 + \sigma\sqrt{t(1 - t)}\mathbf{Z}, \tag{153}$$

where $\mathbf{Z} \sim \mathcal{N}(0, \mathbf{I})$, then the marginal distribution of $\mathbf{R}^t$ becomes $\mathcal{N}((1 - t)z_0 + tz_1, (1 - t)\sigma^2 \mathbf{I})$. We use this distribution to sample geometric state $\mathbf{R}^t$ in the training stage.

**Trajectory guidance.** Similarly, we set $T = \frac{1}{N}$, $p_{\mathcal{R}}^i(z_{i+1}, T|\mathbf{R}^{t'}, t') = \mathcal{N}(\mathbf{R}^{t'}, \sigma_i^2(T - t')\mathbf{I})$ when we use the trajectory guidance. So the h-transformed SDE becomes

$$d\mathbf{R}_i^t = \frac{\mathbf{R}_i^T - \mathbf{R}_i^t}{T - t} dt + \sigma_i d\mathbf{W}^t, \tag{154}$$

which is a Brownian bridge with $T = \frac{1}{N}$. Then associated density function is

$$q_{\mathcal{R}}^i(\mathbf{R}^{t'}, t'|z_{i+1}, T; z_i, 0) = \mathcal{N}(\frac{t'}{T}z_{i+1} + \frac{T - t'}{T}z_i, \sigma_i^2 \frac{t'(T - t')}{T^2}\mathbf{I}). \tag{155}$$

Additionally, we set $\sigma_i$ decays linearly with respect to $\frac{i}{N}$, i.e. $\sigma_i = \frac{N-i}{N}\sigma$, where $\sigma$ is a hyperparameter. Again, in training stage, we set $q_{\mathcal{R}}^i(\mathbf{R}_i^0, 0|z_1, 1; z_0, 0) = \mathcal{N}(z_0, \sigma_i^2 \mathbf{I})$ as initial distribution, and the terminal distribution is $q_{\mathcal{R}}^i(\mathbf{R}_i^0, 0|z_1, 1; z_0, 0) = \mathcal{N}(z_1, \sigma_{i+1}^2 \mathbf{I})$, which is same as the initial distribution of the next bridge.

**Sampling Algorithm** We use the ODE-based method to generate samples at inference time. After the training process, the neural network $\mathbf{v}_\theta$ is trained as described in Algorithm 3 and Algorithm 4. When the network is trained without trajectory guidance, we simulate the following ODE to generate samples:

$$\frac{d\mathbf{R}^t}{dt} = \mathbf{v}_\theta(\mathbf{R}^t, t; \mathbf{R}^0), \mathbf{R}^0 \sim q_{\text{data}}(R^{t_0}), t \in [0, T]. \tag{156}$$

When the network is trained with trajectory guidance, we solve the following ODE to generate samples:

$$\frac{d\mathbf{R}^t}{dt} = \mathbf{v}_\theta(\mathbf{R}^t, t; \mathbf{R}^{\lfloor \frac{t}{T} \rfloor T}), \mathbf{R}^0 \sim q_{\text{data}}(R^{t_0}), t \in [0, N \times T]. \tag{157}$$

Denote a black box ODE solver by $\text{Solver}(\mathbf{v}, t)$. $\text{Solver}(\mathbf{v}, t)$ takes a vector field $\mathbf{v}$ and a time point as inputs, then returns the solution of the ODE

$$\frac{d\mathbf{X}^t}{dt} = \mathbf{v}(\mathbf{X}^t, t; \phi), \mathbf{X}^0 = x_0, \tag{158}$$

at the specific time $t$, i.e. $\text{Solver}(\mathbf{v}, t) = \mathbf{X}^t$. Combining all the above design choices, we have the following algorithms for sampling our Geometric Diffusion Bridge (Algorithm 5) and leveraging trajectory guidance if available (Algorithm 6).

**Algorithm 3** Training

1: **repeat**
2: $\quad (z_0, z_1) \sim q_{\text{data}}(R^{t_0}, R^{t_1})$
3: $\quad t \sim \mathcal{U}[0, T]$
4: $\quad \epsilon \sim \mathcal{N}(\mathbf{0}, \mathbf{I})$
5: $\quad \mathbf{R}^t = \frac{t}{T} z_1 + \frac{T-t}{T} z_0 + \frac{\sqrt{t(T-t)}}{T} \sigma \epsilon$
6: $\quad$ Take gradient descent step on
$$\nabla_\theta \lambda(t) \left\| \frac{z_1 - \mathbf{R}^t}{\sigma^2(T-t)} - \mathbf{v}_\theta(\mathbf{R}^t, t; z_0) \right\|^2$$
7: **until** converged

---

**Algorithm 4** Training with trajectory guidance

1: **repeat**
2: $\quad (z_0, \ldots, z_N) \sim q_{\text{traj}}(\tilde{R}^0, \ldots, \tilde{R}^N)$
3: $\quad t \sim \mathcal{U}(0, N \times T), i = \lfloor \frac{t}{T} \rfloor, t' = t - i \times T$
4: $\quad \epsilon \sim \mathcal{N}(\mathbf{0}, \mathbf{I})$
5: $\quad \mathbf{R}_i^{t'} = \frac{t'}{T} z_{i+1} + \frac{T-t'}{T} z_i + \frac{\sqrt{t'(T-t')}}{T} \sigma_i \epsilon$
6: $\quad$ Take gradient descent step on
$$\nabla_\theta \lambda(t) \left\| \frac{z_{i+1} - \mathbf{R}_i^{t'}}{\sigma_i^2(T-t')} - \mathbf{v}_\theta(\mathbf{R}_i^{t'}, t; z_i) \right\|^2$$
7: **until** converged

---

**Algorithm 5** Sampling

**Require:** Initial geometric state $z_0 \sim q_{\text{data}}(R^{t_0})$, a trained neural network $\mathbf{v}_\theta$, a numerical ODE solver $\text{Solver}(\mathbf{v}, t)$
1: $\mathbf{R}^0 = z_0$
2: $\mathbf{R}^T = \text{Solver}(\mathbf{v}_\theta(\mathbf{R}^t, t; \mathbf{R}^0), T)$
**Ensure:** $\mathbf{R}^T$

---

**Algorithm 6** Sampling with trajectory guidance

**Require:** Initial geometric state $z_0 \sim q_{\text{data}}(R^{t_0})$, a trained neural network $\mathbf{v}_\theta$, a numerical ODE solver $\text{Solver}(\mathbf{v}, t)$
1: $\mathbf{R}^0 = z_0$
2: $\mathbf{R}^{NT} = \text{Solver}(\mathbf{v}_\theta(\mathbf{R}^t, t; \mathbf{R}^{\lfloor \frac{t}{T} \rfloor T}), t = NT)$
**Ensure:** $\mathbf{R}^{NT}$

---

# D  Experiments

## D.1  Equilibrium State Prediction

**Dataset.** QM9 [79] is a quantum chemistry benchmark consisting of 134k stable small organic molecules, which has been widely used for molecular modeling. These molecules correspond to the subset of all 133,885 species out of the GDB-17 chemical universe of 166 billion organic molecules. In convention, 110k, 10k, and 11k molecules are used for train/valid/test sets respectively. The geometric conformations that are minimal in energy are provided in the QM9 dataset. The equilibrium conformation and its relative properties are all calculated at the B3LYP/6-31G(2df,p) level of quantum chemistry.

Molecule3D [116] is a large-scale dataset curated from the PubChemQC project [67, 71], consisting of 3,899,647 molecules in total, 2,339,788 molecules in training set, 779,929 molecules in the validation set, 779,930 molecules in the test set, and its train/valid/test splitting ratio is $6 : 2 : 2$. For each molecule, the 2D atom graph, the 3D equilibrium geometric conformation, and four extra properties are provided. In particular, both random and scaffold splitting methods are adopted to thoroughly evaluate the in-distribution and out-of-distribution performance. For each molecule, an initial geometric state is generated by using fast and coarse force field [73, 52] and geometry optimization is conducted to obtain B3LYP/6-31G* level DFT-calculated equilibrium geometric structure.

**Baselines.** We comprehensively compare our GDB framework with previous equilibrium conformation prediction methods. Following [111], we use DG and ETKDG algorithms implemented by RDkit as our fundamental baselines. The benchmark [116] used the DeeperGCN-DAGNN framework [60] which proposed a deep graph neural network architecture to predict 3D geometric conformation of the molecule based on its 2D graph structure, and got impressive performance on the Molecule3D dataset. GINE [39] proposed a method for pretraining GNN to improve the performance and capacity of GNN. GATv2 [10] proposed a dynamic graph attention mechanism and improved the performance of the graph attention network on several tasks. GPS [80] proposed a general framework that supported multiple types of encodings with efficiency and scalability guarantees in both small and large graph prediction tasks. GTMGC [111] proposed a novel neural network based on Graph-Transformer (GT) [118, 66, 119, 65] to predict the equilibrium conformation of the molecule in 3D based on its 2D graph structure.

**Metric.** Following [116], three metrics are adopted to evaluate predictions of equilibrium states: (1) C-RMSD: given prediction $\hat{R} = \{\hat{\mathbf{r}}_i\}_{i=1}^N$ which is rigidly aligned to the ground-truth $R^* = \{\mathbf{r}_i^*\}_{i=1}^N$ by the Kabsch algorithm [44], Root Mean Square Deviation between their atoms is calculated, i.e., $\text{C-RMSD}(\hat{R}, R^*) = \sqrt{\frac{1}{N} \sum_{i=1}^N \|\hat{\mathbf{r}}_i - \mathbf{r}_i^*\|_2^2}$; (2) D-RMSE: based on $\hat{R}$ and $R^* = \{\mathbf{r}_i^*\}_{i=1}^N$, interatomic distances can be calculated, i.e., $\{\hat{d}_i\}_{i=1}^{N'}$ and $\{\hat{d}_i^*\}_{i=1}^{N'}$. Root Mean Square Error be-

tween these distances is calculated, i.e., D-RMSE$(\{\hat{d}_i\}_{i=1}^{N'}, \{\hat{d}_i^*\}_{i=1}^{N'}) = \sqrt{\frac{1}{N'}\sum_{i=1}^{N}(d_i - d_i^*)^2}$; (3) D-MAE$(\{\hat{d}_i\}_{i=1}^{N'}, \{\hat{d}_i^*\}_{i=1}^{N'}) = \frac{1}{N'}\sum_{i=1}^{N}|d_i - d_i^*|$.

**Settings.** In this task, we parameterize $\mathbf{v}_\theta(\mathbf{R}^t, t; \mathbf{R}^0)$ by extending a Graph-Transformer based equivariant network [92, 63] to encode both time steps and initial geometric states as conditions. For training, we use AdamW as the optimizer, and set the hyper-parameter $\epsilon$ to 1e-8 and $(\beta_1, \beta_2)$ to (0.9,0.999). The gradient clip norm is set to 5.0. The peak learning rate is set to 1e-4. The batch size is set to 512. The weight decay is set to 0.0. The model is trained for 500k steps with a 30k-step warm-up stage. After the warm-up stage, the learning rate decays linearly to zero. The noise scale $\sigma$ is set to 0.5. For inference, we use 10 time steps with the Euler solver [12]. All models are trained on 16 NVIDIA V100 GPU.

## D.2 Structure Relaxation

**Dataset.** Open Catalyst 2022 (OC22) dataset [105] is a widely used dasaset, which has great significance for the development of Oxygen Evolution Reaction (OER) catalysts. Each data in the dataset is in the form of the adsorbate-catalyst complex. Both initial and adsorption states with trajectories connecting them are provided. The dataset consists of 62,331 Density Functional Theory (DFT) relaxations trajectories, and about 9,854,504 single-point DFT calculations across a range of oxide materials, coverages, and adsorbates.The training set consists of 45,890 catalyst-adsorbate complexes. To better evaluate the model's performance, the validation and test sets consider the in-distribution (ID) and out-of-distribution (OOD) settings which use unseen catalysts, containing approximately 2,624 and 2,780 complexes respectively.

**Baselines.** Following [105], we choose strong MLFF baselines trained on force field data for a challenging comparison. Spinconv [94] introduced a novel approach called spin convolution to model angular information between sets of neighboring atoms in a graph neural network and got impressive performance in molecular simulation tasks. Gemnet [32] proposed multiple structural improvements for geometric GNN with theoretical insights, which significantly improved the experimental performance as well. Based on Gemnet's framework, Gemnet-OC [34] modified the architecture of the network and improved the experimental performance on more diverse tasks.

In [105], there are still other baseline setting. [105] introduce a large-scale dataset Open Catalyst 2020 (OC20), which consists of 1,281,040 Density Functional Theory (DFT) relaxations and 264,890,000 single point evaluations to help training the baseline model. [105] presented baselines using both OC20 and OC22 data in training stage and baselines using only OC20/OC22 for comparison.

**Metric.** Following [105], we use the Average Distance within Threshold (ADwT) as the evaluation metric, which reflects the percentage of structures with an atom position MAE below thresholds. To be more precise, the ADWT metric across thresholds ranging from $\beta = 0.01\mathring{A}$ to $\beta = 0.5\mathring{A}$ in increments of $0.001\mathring{A}$. The computation of ADwT metric is to count the percentage of structures with an atom position MAE below the threshold.

**Settings.** In this task, We parameterize $\mathbf{v}_\theta(\mathbf{R}^t, t; \mathbf{R}^0)$ by using GemNet-OC [34], which also serves as a verification that our framework is compatible with different backbone models. For training, we use AdamW as the optimizer, and set the hyper-parameter $\epsilon$ to 1e-8 and $(\beta_1, \beta_2)$ to (0.9,0.999). The gradient clip norm is set to 10.0. The peak learning rate is set to 5e-4. The batch size is set to 64. The weight decay is set to 0.0. The model is trained for 200k steps. After the warm-up stage, the learning rate decays linearly to zero. The noise scale $\sigma$ is set to 0.5. The trajectory length is set to $N = 10$. For inference, we also use 10 time steps with the Euler solver [12]. All models are trained on 8 NVIDIA A100 GPU.

