# OpenReview forum: "Bridging Geometric States via Geometric Diffusion Bridge"
_NeurIPS.cc/2024/Conference — NeurIPS 2024 poster_

### Official Review · Reviewer_dV6w · 2024-07-12

**Soundness:** 3
**Presentation:** 4
**Contribution:** 3
**Rating:** 6
**Confidence:** 3

**Summary:**

The paper introduces the Geometric Diffusion Bridge (GDB), a novel framework designed to generate the evolution of geometric states in geometric (coordinate) systems. GDB uses a diffusion bridge connecting initial and target geometric states with equivariant transition kernels, preserving symmetry and joint state distributions. Furthermore, GDB can use a chain of equivariant diffusion bridges to leverage trajectory data for more accurate dynamic modeling.

**Strengths:**

- The presentation of theorems in Section 3.1 is clear and straightforward, establishing a solid theoretical foundation for GDB. The authors effectively derive theorems and integrate them with point cloud states.
- GDB demonstrates strong performance across various tasks, including QM9, Molecule3D, and OpenCatalyst IS2RS.

**Weaknesses:**

I have no complaints regarding the technical and experimental sections, as they are well-written. However, I wonder existing works, such as [1] and [2], also use diffusion bridges over molecular data. What advantages does your approach have over theirs?

[1] Diffusion-based Molecule Generation with Informative Prior Bridges. Lemeng Wu, et al. NeurIPS 2022.

[2] DiSCO: Diffusion Schrödinger Bridge for Molecular Conformer Optimization. Danyeong Lee, et al. AAAI 2024.

**Questions:**

See weaknesses.

**Limitations:**

The authors note the need of exploring better implementation strategies for their framework to enhance performance.

---

> ### Author Rebuttal · Authors · 2024-08-07
>
> Thank you for recognizing both the theoretical analysis and practical effectiveness of our GDB framework. We also appreciate your suggestions which can improve our work further.
>
> Our proposed method has the following advantages compared to your list of works [1, 2].
>
> - First, our proposed method can make good use of trajectory data during training (trajectory guidance), which [1, 2] cannot. Trajectory data provide valuable insights into the evolution of geometric states. We conducted a theoretical analysis, which showed that our approach can preserve the joint distribution of the trajectory (Theorem 3.4) with strong expressiveness guarantees (Theorem 3.5). Empirical results on the large-scale real-world benchmark OC22 also verify the superiority of trajectory guidance enabled by our framework.
>
> - Second, our GDB framework can preserve the coupling of geometric states, which is also crucial and necessary in modeling the evolution of geometric states (please refer to line 132 in section 3 and Theorem 3.3 for more details).  Diffusion bridge works the reviewer mentioned [1, 2] do not guarantee coupling preservation. Our paper provides detailed derivations and theoretical analysis for the coupling preservation of geometric states, which we believe add value to the communities of both geometric deep learning and diffusion bridge-based approaches.
>
> Besides these contributions, we also provide detailed derivations and theoretical guarantees for satisfying the symmetry constraints of the equivariant diffusion bridge. Although both the equivariant diffusion process and diffusion bridge have been studied, few works exist to thoroughly introduce methodologies for combining the power from both sides, while our work makes it complete.
>
> We will carefully cite the listed works and add the above discussions to the next version of our paper to let the general audience better understand our contributions.
>
> [1] Diffusion-based Molecule Generation with Informative Prior Bridges. Lemeng Wu, et al. NeurIPS 2022.
>
> [2] DiSCO: Diffusion Schrödinger Bridge for Molecular Conformer Optimization. Danyeong Lee, et al. AAAI 2024.
>
> We thank you again for your efforts in reviewing our paper, and we have replied to each of your comments. We look forward to your re-evaluation of our submission based on our responses and updated results.

---

> ### Author Response · Authors · 2024-08-10
> **Looking forward to your re-evaluation**
>
> Dear Reviewer dV6w,
>
> Thank you for your time and efforts in reviewing our paper. We have carefully responded to each of your questions. Given that the author-reviewer discussion deadline is approaching, we would greatly appreciate it if you could kindly take a look at our responses and provide your valuable feedback. We are more than happy to discuss more if you still have any concerns.
>
> Thank you once again and we are eagerly looking forward to your re-evaluation of our work.
>
> Paper 15795 Authors

---

> ### Comment · Reviewer_dV6w · 2024-08-12
> **Official Comment by Reviewer dV6w**
>
> Thank you for your reply and I am satisfied with the responses. I will keep my positive score.

---

> > ### Author Response · Authors · 2024-08-13
> >
> > Thanks for your reply. We are happy to hear that all your concerns have been properly addressed. We thank you again for recognizing our contributions and staying positive about our work.

---

### Official Review · Reviewer_U4NU · 2024-07-13

**Soundness:** 3
**Presentation:** 3
**Contribution:** 2
**Rating:** 5
**Confidence:** 3

**Summary:**

This paper proposes a generative model for bridging initial and target geometric states using diffusion bridge. This work introduces an equivariant diffusion bridge based on equivariant transition kernels for symmetry constraints. The proposed method was validated on diverse settings including simple molecules and adsorbate-catalyst complex, outperforming previous MLFF baselines.

**Strengths:**

- The motivation of using diffusion bridge to bridge initial and target geometrical states is reasonable.
- Using diffusion bridge model for equilibrium state prediction and structure relaxation is novel to the best of my knowledge, and the paper shows that GDB significantly outperforms previous methods with diverse datasets.
- Equivariant design of bridge process is based on solid theory.
- The paper is well written except for some missing relevant works on diffusion bridge.

**Weaknesses:**

- Related works on diffusion bridges or diffusion mixtures were not discussed. Diffusion bridges has been studied in [1,2,3,4] with applications to molecules, graphs, point clouds, and images, and more recent works have studied general framework for diffusion bridges [5, 6] which is worth discussing. While GDB has a contribution for using diffusion bridges in new tasks, discussing related works and clarifying the novel contributions is necessary in particular for strengthening the contribution of this work.
- Contribution seems limited as using diffusion bridge as generative modeling was already studied [1,2,3,4], in particular deriving diffusion bridges using Doob's h-transform. Designing an equivariant diffusion process (not necessarily bridge) specifically in SE(3) group has been covered in [7,8, 9]. What is the difference of designing equivariant diffusion bridges compared to equivariant diffusion processes?

[1] Peluchetti, Diffusion Bridge Mixture Transports, Schrodinger Bridge Problems and Generative Modeling, JMLR 2023
[2] Liu et al., Learning Diffusion Bridges on Constrained Domains, ICLR 2023
[3] Wu et al., Diffusion-based Molecule Generation with Informative Prior Bridges, NeurIPS 2022
[4] Jo et al., Graph Generation with Destination-Predicting Diffusion Mixture, arXiv 2023
[5] Albergo et al., Stochastic Interpolants: A Unifying Framework for Flows and Diffusions, arXiv 2023
[6] Shi et al., Diffusion Schrodinger Bridge Matching, NeurIPS 2023
[7] Xu et al., GeoDiff: a Geometric Diffusion Model for Molecular Conformation Generation, ICLR 2022
[8] Xu et al., Geometric Latent Diffusion Models for 3D Molecule Generation, ICML 2023
[9] Yim et al., SE(3) diffusion model with application to protein backbone generation, ICML 2023

**Questions:**

- What is the reason for using deterministic process (i.e., probability flow ODE) instead of the original stochastic process? Does ODE results in better performance?
- Is GDB scalable to geometric states of high dimensions? While analysis on this may not be necessary, it could strengthen the work.

**Limitations:**

While the paper discusses future direction for the proposed method, specific limitations of the work is not specified. One potential issue might the scalability of GDB as the model has transformer architecture, and other issue could be long inference time which is a typical problem of diffusion models.

---

> ### Author Rebuttal · Authors · 2024-08-07
>
> Thank you for recognizing the motivation, contributions, and theoretical analysis of our GDB framework. We also appreciate your suggestions which can improve our work further. Here are our responses to your questions.
>
> >**Regarding discussions of related works.**
>
> Thank you for listing these related works [1-6] on diffusion bridges. We agree that adding more discussions on these related works would help the audience to understand our contributions better. We will carefully cite all these works, compare them with our work regarding their contributions, and update these discussions in the next version of our paper.
>
> >**Regarding our contributions.**
>
> Our work cannot be considered a simple extension or follow-up work of constructing diffusion bridges with SE(3) symmetry constraint ([1-4,7-9]). As described throughout the whole manuscript, the task we tackle is to capture the evolution of geometric states. This task has unique difficulties compared to conventional generative models, including **how to leverage trajectory data and preserve coupling between geometric states over time**. Simply combining existing diffusion bridge approaches with equivariant diffusion processes cannot meet the requirement.
>
> To the best of our knowledge, our GDB framework is the first approach to leverage the characteristics of diffusion bridges for flexibly incorporating trajectory guidance. We conduct theoretical analysis to show that our approach can preserve the joint distribution of the trajectory (Theorem 3.4) with strong expressiveness guarantees (Theorem 3.5). Empirical results on the large-scale real-world benchmark OC22 also verify the superiority of trajectory guidance enabled by our framework.
>
> Moreover, our GDB framework preserves the coupling of geometric states, which is also crucial and necessary in modeling the evolution of geometric states (please refer to line 132 in section 3 and Theorem 3.3 for more details). SE(3) diffusion models [7,8,9] are restricted to transport from the standard Gaussian distribution to the target distribution of molecules or proteins, which are obviously unable to preserve the coupling of geometric states. Previous works on diffusion bridges [1,2,3,4,5,6], which the reviewer has mentioned, do not provide guarantees on coupling preservation, especially for distributions of geometric states. Our paper provides detailed derivations and theoretical analysis for the coupling preservation of geometric states, which we believe add value to the communities of both geometric deep learning and diffusion bridge-based approaches.
>
> We will carefully add the above discussions to the next version of our paper to let the general audience better understand our contributions.
>
> >**Regarding the usage of the ODE sampler.**
>
> As stated in lines 259-265 in our paper, we leverage ODE solvers for inference due to efficiency considerations. This is motivated by [10], which has also become a common choice in the literature of diffusion models for efficiency sampling. Our framework can also be implemented by using other advanced sampling strategies, which we leave as future works.
>
> [10]. Song, Yang, et al. "Score-based generative modeling through stochastic differential equations." ICLR 2021.
>
> >**Regarding the scalability of GDB to geometric states of high dimensions.**
>
> Thank you for the suggestion. In Table 2 of our paper, we briefly introduce the types and scales of datasets we used, covering both medium-scale and large-scale real-world benchmarks. Here we present more details on the data dimensions of these benchmarks:
>
> - QM9 is a medium-scale dataset that consists of \~130,000 organic molecules (see line 277 in section 4.1). Molecules in this dataset typically consist of 9 heavy atoms C,O,N,F (not counting hydrogens), so the dimension of geometric states is $9\times 3 = 27$.
>
> - Molecule3D is a large-scale dataset curated from the PubChemQC project, consisting of 3,899,647 molecules in total (see line 279 in section 4.1). In the Molecule3D dataset, each molecule includes 29.11 atoms on average. The dimension of geometric states is around 87 on average.
>
> - The Open Catalyst 2022 (OC22) dataset consists of 62,331 Density Functional Theory (DFT) relaxations, which have great significance for the development of Oxygen Evolution Reaction (OER) catalysts. Each data is in the form of the adsorbate-catalyst complex (see line 305 in section 4.2 for more details), consisting of hundred-scale atoms with periodic boundary conditions being considered. The dimension of geometric states in this dataset is thus in hundred to thousand scales.
>
> From the above statistics, we can see that our GDB framework is general to be applied to accurately predict the evolution of both low-dimensional and high-dimensional geometric states. Bridging geometric states of higher dimensions, such as protein conformation states, indeed has great significance in various scientific problems like transition state discovery. In our ongoing work, we successfully extend our GDB framework to this higher-dimensional problem with satisfactory preliminary results, which demonstrate the promising scalability of GDB for bridging geometric states of higher dimensions.
>
> >**Regarding the discussion of limitations**
>
> Thank you for the suggestion. For the sake of generality, we do not experiment with advanced implementation strategies of training objectives and sampling algorithms, which leave room for further improvement. Besides, the employment of Transformer-based architectures may also limit the efficiency of our framework. This has also become a common issue in transformer-based diffusion models. We will organize these discussions into a single Limitation section and update it in the next version of our paper.
>
> We thank you again for your efforts in reviewing our paper, and we have replied to each of your comments. We look forward to your re-evaluation of our submission based on our responses and updated results.

---

> ### Author Response · Authors · 2024-08-10
> **Looking forward to your re-evaluation**
>
> Dear Reviewer U4NU,
>
> Thank you for your time and efforts in reviewing our paper. We have carefully responded to each of your questions. Given that the author-reviewer discussion deadline is approaching, we would greatly appreciate it if you could kindly take a look at our responses and provide your valuable feedback. We are more than happy to discuss more if you still have any concerns.
>
> Thank you once again and we are eagerly looking forward to your re-evaluation of our work.
>
> Paper 15795 Authors

---

> > ### Author Response · Authors · 2024-08-13
> > **Kindly request for feedback and reevaluation**
> >
> > Dear Reviewer U4NU,
> >
> > Thank you once again for taking the time to review our paper! As the Reviewer-Author discussion deadline is quickly approaching, we would sincerely appreciate it if you could provide us with further feedback on our responses and kindly reevaluate our work based on our clarification and additional results in our rebuttal.
> >
> > Following your insightful suggestions, we have thoroughly discussed the related works in the rebuttal and clarified our novel contributions. We also comprehensively illustrate GDB's scalability to geometric states of high dimensions. Additionally, we provide additional details and our motivation for the ODE sampler. Based on these additional results and clarifications, we sincerely hope your concerns can be addressed.
> >
> > We always believe that the feedback between reviewers and authors would indeed improve the paper's quality, and we will definitely include the related works discussions from the reviewer's suggestions in the revised paper. It would be really nice to see both of us reach a consensus. We sincerely look forward to your reevaluation and feedback!
> >
> > Best Regards,
> >
> > Paper 15795 Authors

---

> ### Author Response · Authors · 2024-08-14
> **Kindly Reminder of the Close of the Discussion Period**
>
> Dear Reviewer U4NU,
>
> We would like to express our gratitude for your valuable comments and feedback. As the author-reviewer discussion period is coming to close on Aug 13th, we would greatly appreciate it if you could provide more feedback and reevaluate our work based on our responses and updated results. Thank you very much for your attention to this matter.
>
> Best regards,
>
> Authors

---

### Official Review · Reviewer_znC7 · 2024-07-14

**Soundness:** 2
**Presentation:** 2
**Contribution:** 2
**Rating:** 3
**Confidence:** 4

**Summary:**

This paper proposes a type of diffusion model that captures the evolution of geometric states. The model is characterized by a diffusion SDE that couples the initial state with the target state, in the middle of which trajectory guidance is enabled when such data present. The framework is designed to yield equivariant density similar to other geometric diffusion models. Experiments on equilibrium state prediction with or without trajectory data have been performed to verify the applicability of the proposed approach.

**Strengths:**

1. The distinction between existing works has been elaborated in Table 1, which is clear.

2. The method is designed with an option to leverage additional trajectory data, which is quite interesting.

**Weaknesses:**

1. The experimental setup and comparison with baselines on equilibrium state prediction is a bit troublesome which requires more clarification or additional comparisons. Please refer to Q1.

2. The presentation is a bit unclear. Please refer to Q2.

3. Additional baselines may be considered. The baselines selected in the paper are not closely connected to the proposed approach. See Q3.

4. Missing ablation studies. In the current shape it is unclear where the performance gain comes from. See Q4.

**Questions:**

Q1. The evaluation protocol on QM9 and Molecule3D, especially to compare with direct prediction approaches, is not a common practice. A more convincing benchmark protocol would be to compare with methods such as GeoDiff [1] on molecule generation tasks since they are also generative models. Since the paper is positioned to tackle generative modeling, the experiments should also be designed to align with the goal.

Q2. Could the authors provided detailed sampling algorithm this approach adopts? If the model uses sampling approach similar to other diffusion models, there should be related discussions on sampling steps/sampling time the method consumes.

Q3. A more reasonable baseline would be to directly apply existing bridge models (e.g., [2]) to the current task by switching the backbone to the one this paper adopts. This would help the audience understand the unique contribution of this work since both bridge models and equivariant (geometric) diffusion models have been proposed in literature.

Q4. Ablation studies such as investigating the importance of preserving equivariance of the density modeled should be included. This would help justify the necessity of the proposed components.


[1] Xu et al. GeoDiff: a Geometric Diffusion Model for Molecular Conformation Generation. In ICLR'22.

[2] Zhou et al. Denoising Diffusion Bridge Models. In ICLR'24.

**Limitations:**

There seem to be no discussions on limitations in the paper. It would be better to discuss potential limitations from perspectives such as scalability and sampling time.

---

> ### Author Rebuttal · Authors · 2024-08-07
>
> Thank you for spending time reviewing our paper. We would like to first address your misunderstanding by **clarifying the task of our interest**: to capture/predict the evolution of geometric states, i.e., *predicting future states from initial states*. This goal has been carefully stated at the very beginning of our paper (please refer to (1) line 1 in Abstract; (2) line 22 in Sec 1 Introduction; (3) lines 83-84 in Sec 2.1 Problem Definition). We also provide its formal definition. See lines 84-91.
>
> Based on these statements, we would like to argue that **our scope is different from conventional molecular generation problems**, such as the one introduced in GeoDiff mentioned in the review comments. In Sec 3.1 of GeoDiff, the definition of molecule generation task is "Given multiple **graphs $G$**, and for each $G$ given **a set of** conformations $C$, ..., the goal is learning a generative model $p_\theta(C|G)$". From the definition, we can see:
>
> - The input in GeoDiff is the molecular graph, while in ours, the input is the geometric state of a system at a given time $t_0$.
>
> - The output of the molecule generation task is a set of sampled conformations $C$ from Boltzmann distribution, while the output of our task is the geometric state at a given time $t_1$ evolved from time $t_0$.
>
> - The goal of GeoDiff is to learn a generative model for the underlying Boltzmann distribution of molecular conformations, while our task is to capture the evolution of geometric states over time.
>
> Due to these differences, our task has **several unique difficulties, including how to leverage trajectory data and preserve coupling between states over time while satisfying SE(3) at the same time**. Our work tackled these challenges and provided promising results.
>
> We guess that the misunderstanding may partly be due to the lack of detailed task descriptions for the related works. We hope the above clarifications can help the reviewer better understand our task definition and our goal. We will update these discussions in the next version of our paper to avoid ambiguity and misunderstanding.
>
> >**Regarding the experimental setup and evaluation protocol (Weakness 1 & Q1)**
>
> **First, we believe our evaluation protocol aligns well with our task**. To measure how accurate a model is in predicting a system's future geometric state from its initial geometric state, we follow existing works in this direction [1, 2, 3] to use C-RMSD, D-MAE, and D-RMSE on QM9/Molecule3D and ADwT on OC22 which comprehensively measure the position/distance error between predicted geometric state and ground-truth geometric state.
>
> The reviewer suggested that we should follow GeoDiff and use their proposed metric. We are afraid that it is not a proper choice. GeoDiff (In Sec.5.2) calculates metrics between two **sets** of conformations, which aims at measuring the difference between the learned distribution and ground truth distribution rather than how accurate a target geometric state is predicted when specifying the initial geometric state.
>
> **Second, we believe strong and latest baselines in this direction have been included**. We followed the most recent works (e.g., GTMGC in ICLR24) and benchmarks (e.g., OC22) to compare our approach. The reviewer suggested that we should use GeoDiff as the baseline. But again, it is not a proper choice as GeoDiff cannot leverage trajectory guidance if provided and cannot take the initial geometric state as input to predict the target geometric state, which has been clearly discussed at the beginning of our response.
>
> >**Regarding the details of our sampling algorithm (Weakness 2 & Q2)**
>
> We have already discussed the sampling/inference method (ODE solver) of our GDB framework in lines 261-265 and presented details of the ODE solver (here we use the basic Euler solver) and sampling steps (10 steps in total) in Appendix C. Following your suggestion, we will provide an additional pseudo-code of our inference process in the next version of our paper.
>
> >**Regarding additional baseline comparisons and the unique contribution of our work (Weakness 3 & Q3)**
>
> First, we would like to clarify the reviewer's concern: "The baselines selected in the paper are not closely connected to the proposed approach." As stated in our responses to Weakness 1 & Q1, our task of interest is to predict the future geometric state of a system from its initial geometric state. In both Sec 1 Introduction and Sec 5 Related Works, we carefully review existing approaches for this task, including experimental approaches, traditional computational methods, direct prediction, and machine learning force fields. Although "both bridge models and equivariant (geometric) diffusion models have been proposed in literature" as the reviewer mentioned, there exist few works using generative modeling techniques to comprehensively investigate our task of interest due to the difficulty of unified satisfying the three desiderata (Coupling Preservation, Symmetry Constraints and Trajectory Guidance, lines 132-145). Following existing works, we use strong baselines of this task for comparisons, which is common practice.
>
> Following your suggestion, we compare our approach with Denoising Diffusion Bridge Models by switching the backbones on OC22. Our GDB framework significantly outperforms this new baseline by 6.37% on average ADwT. Moreover, our approach can further leverage trajectory guidance to achieve better performance, i.e., 7.93% improvement compared to this baseline. We will update these results in the next version of our paper, which we believe can help the audience better capture our GDB framework's effectiveness.

---

> ### Author Response · Authors · 2024-08-07
> **Rebuttal by Authors (Part 2)**
>
> >**Regarding ablation studies (Weakness 4 & Q4)**
>
> We have already provided ablation studies. Please refer to Sec 4.2 (lines 335-342 and Table 5) for detailed descriptions and results. Our ablation studies help us better understand the importance of key designs in our GDB framework, including trajectory guidance and coupling preservation, without which we observe significant performance drops. Following your suggestion, we further conduct an ablation study by investigating the importance of satisfying equivariant constraints, which shows that the performance drop is 14.45% if we remove the equivariant constraints. These ablation studies serve as supporting evidence for the necessity of proposed components in our GDB framework, and we will update these results in the next version of our paper.
>
> >**Regarding the discussion of limitations**
>
> Thank you for the suggestion. For the sake of generality, we do not experiment with advanced implementation strategies of training objectives and sampling algorithms, which leave room for further improvement. Besides, the employment of Transformer-based architectures may also limit the efficiency of our framework. This has also become a common issue in transformer-based diffusion models. We will organize these discussions into a single Limitation section and update it in the next version of our paper.
>
> We thank you again for your efforts in reviewing our paper, and we have replied to each of your concerns. We sincerely look forward to your re-evaluation of our submission based on our responses and updated results.

---

> ### Author Response · Authors · 2024-08-10
> **Looking forward to your re-evaluation**
>
> Dear Reviewer znC7,
>
> Thank you for your time and efforts in reviewing our paper. We have carefully responded to each of your questions. Given that the author-reviewer discussion deadline is approaching, we would greatly appreciate it if you could kindly take a look at our responses and provide your valuable feedback. We are more than happy to discuss more if you still have any concerns.
>
> Thank you once again and we are eagerly looking forward to your re-evaluation of our work.
>
> Paper 15795 Authors

---

> ### Comment · Reviewer_znC7 · 2024-08-11
>
> I thank the authors for the response. However, some concerns are still not addressed.
>
> While the original setting in GeoDiff is molecule generation, from my perspective, it could be easily extended to your setting with the core difference lying in tackling how to condition on an input structure, which can be tackled in a fairly easy way. A related approach that considers such extension is DiffMD [1]. A strong reason why these approaches are compelling is that your approach is claimed to be generative, in which case some generative baselines (or even benchmarks) should be included.
>
> [1] Wu et al. DiffMD: A Geometric Diffusion Model for Molecular Dynamics Simulations. In AAAI 2023.
>
> Regarding Q2, to me it is indeed quite surprising that only 10 steps are needed to obtain high quality samples, as opposed to 100-1000 steps adopted in previous works. Thus it would be interesting to see how the performance changes with the number of sampling steps.

---

> > ### Author Response · Authors · 2024-08-12
> >
> > Thanks for your quick reply and constructive suggestions. We agree with your point that including more generative baselines for comparisons could improve our work. In previous responses, we followed your advice to include Denoising Diffusion Bridge Models with our equivariant backbone, which combines advanced bridge models and equivariant models as a strong generative baseline. We will also follow your suggestion to include DiffMD in our work. However, we have tried our best but found that DiffMD is not open-sourced. We have to reimplement the method and conduct the experiment, which cannot be finished before the deadline. We will definitely update the results in the next version of our paper. We will also carefully cite these generative baselines and provide thorough discussions in our Related Work section.
> >
> > We use 10-step sampling for the sake of simplicity. In our preliminary experiments, we found that using ten steps already yields good performance. Increasing inference steps still improves model performance slightly but with more computational cost. We believe investigating advanced sampling strategies for our method is important, which we leave as future work.
> >
> > Through intensive and meaningful discussions with you, we have realized that some terminology could be more precise. For instance, the title '...via generative modeling' is too broad and might give the impression that we are applying conventional generative modeling methods on a standard generative task. **As discussed in the thread, this is not our intention**. We will clarify these parts to reflect our actual focus better.
> >
> > Thank you again for your valuable suggestions. We sincerely hope the reviewer re-evaluates our work based on these responses and updated results for the next version of our paper.

---

> > > ### Author Response · Authors · 2024-08-13
> > > **Kindly request for feedback and reevaluation**
> > >
> > > Dear Reviewer znC7,
> > >
> > > Thank you once again for taking the time to review our paper! As the deadline for Reviewer-Author discussions is fast approaching, we would greatly appreciate it if you could kindly reevaluate our work with updated scores. Overall, we have carefully put every effort to address the reviewer's mentioned concerns. Besides, we will definitely follow your suggestion to include updated baseline results and thorough discussions of your mentioned related works in our next version of the paper, which we believe can improve the quality of our submission and address your remaining concerns.
> > >
> > > We always believe that the feedback between reviewers and authors would indeed improve the paper's quality, and we will definitely include the updated results and conceptual discussions from the reviewer's suggestions in the revised paper. It would be really nice to see both of us reach a consensus. We sincerely look forward to your reevaluation and feedback!
> > >
> > > Best Regards,
> > >
> > > Paper 15795 Authors

---

> ### Author Response · Authors · 2024-08-14
> **Kindly Reminder of the Close of the Discussion Period**
>
> Dear Reviewer znC7,
>
> We would like to express our gratitude for your valuable comments and feedback. As the author-reviewer discussion period is coming to close on Aug 13th, we would greatly appreciate it if you could provide more feedback and reevaluate our work based on our responses and updated results. Thank you very much for your attention to this matter.
>
> Best regards,
>
> Authors

---

### Official Review · Reviewer_Sjdg · 2024-07-24

**Soundness:** 3
**Presentation:** 3
**Contribution:** 3
**Rating:** 6
**Confidence:** 2

**Summary:**

In this paper, the authors introduce a Geometric Diffusion Bridge (GDB) framework, which aims to predict the evolution of geometric states in complex systems accurately, crucial for fields such as quantum chemistry and material modeling. Traditional methods face computational challenges, while deep learning approaches lack precision and generality. The authors use Doob’s h-transform to construct an equivariant diffusion bridge. By applying Doob’s h-transform, the authors adjust the SDE to ensure that the process starts from an initial geometric state and is conditioned to reach a target geometric state. This ensures that the transformed process respects the symmetry constraints of the geometric states, leading to more accurate and physically meaningful predictions.

**Strengths:**

+ The framework utilizes an equivariant diffusion bridge derived from a modified Doob’s h-transform. This ensures that the diffusion process respects symmetry constraints, making the predictions more robust and reliable.
+ The paper provides a theoretical framework analysis about preserving symmetries and accurately modeling evolution dynamics.
+ Experimental evaluations show that GDB is better than state-of-the-art approaches in various real-world scenarios, including equilibrium state prediction and structure relaxation tasks.
+ The framework achieves significant error reduction compared to strong baseline models, particularly in challenging tasks such as structure relaxation in the Open Catalyst 2022 dataset

**Weaknesses:**

- The framework, especially when leveraging trajectory data, might introduce significant computational overhead. The simulation-free matching objective is designed to be efficient, but the overall framework’s computational demands might still be high
- Some mathematical notations and definitions in the paper could be made clearer. For instance, explicitly defining all variables and functions used in the modified Doob’s h-transform and constructing equivariant diffusion bridges would improve readability and understanding.

**Questions:**

See above

**Limitations:**

No limitations are addressed in the paper by the authors

---

> ### Author Rebuttal · Authors · 2024-08-07
>
> Thank you for recognizing both the theoretical analysis and practical effectiveness of our GDB framework. We also appreciate your suggestions which can improve our work further. Here are our responses to your questions.
>
> >**Regarding the computational cost of our GDB framework**
>
> Thanks for the question. Indeed, our GDB framework is efficient. Here, we discuss the reasons in the following two settings:
>
> - Setting 1: Bridging geometric states **without trajectory guidance**. In this setting, our training objective (Equ.(5) and Algorithm 1) and inference process are similar to typical diffusion models, which do not bring additional computational overhead.
>
> - Setting 2: Bridging geometric states **with trajectory guidance**. In this setting, trajectory data is used during training. From Equ.(6) and Algorithm 2 in our paper, it can be seen that the loss calculation at each iteration is still efficient (compared to Algorithm 1) with no computational overhead. For inference, we simulate the ODE in time interval $[0, NT]$. In practice, we set $T=1/N$ so the time interval is still $[0,1]$, which is kept the same as setting 1. The intuition is that we split the total time interval $[0,1]$ into $N$ parts for our Chain of Equivariant Diffusion Bridges with $N$ chains. Since the total length of the time interval is the same, predicting the future geometric state from the initial geometric state by using our model trained with trajectory guidance does not bring significant computational overhead.
>
> We find your question highlights the need for a detailed table to introduce the inference algorithm. We will incorporate this in the next version. We hope this addition clarifies the efficiency of our framework and highlights the improvement in model performance.
>
> >**Regarding the mathematical notations and definitions**
>
> Thanks for your suggestion of explicitly defining all variables and functions. We will revise our paper and add a notation paragraph to elaborate on all mentioned notations and concepts in Sections 3.1 & 3.2 for improved clarity and readability.
>
> We thank you again for your efforts in reviewing our paper, and we have replied to each of your concerns. We sincerely look forward to your re-evaluation of our submission based on our responses.

---

> ### Author Response · Authors · 2024-08-10
> **Looking forward to your re-evaluation**
>
> Dear Reviewer Sjdg,
>
> Thank you for your time and efforts in reviewing our paper. We have carefully responded to each of your questions. Given that the author-reviewer discussion deadline is approaching, we would greatly appreciate it if you could kindly take a look at our responses and provide your valuable feedback. We are more than happy to discuss more if you still have any concerns.
>
> Thank you once again and we are eagerly looking forward to your re-evaluation of our work.
>
> Paper 15795 Authors

---

> > ### Comment · Reviewer_Sjdg · 2024-08-11
> >
> > I have read the comments and am satisfied with the responses. However, I would still stick to my original score.

---

> > > ### Author Response · Authors · 2024-08-12
> > >
> > > Thanks for your reply. We are happy to hear that all your concerns have been properly addressed. We thank you again for recognizing our contributions and staying positive about our work.

---

### Decision · Program_Chairs · 2024-09-25

**Decision:**

Accept (poster)

**Comment:**

The paper introduces the Geometric Diffusion Bridge (GDB), a novel framework for predicting the evolution of geometric states in complex systems using an equivariant diffusion bridge approach. The reviewers generally acknowledge the technical soundness and the innovative aspects of the proposed method, particularly its ability to preserve symmetry constraints and leverage trajectory data for enhanced dynamic modeling. Reviewer Sjdg highlights that “GDB surpasses existing state-of-the-art approaches in various real-world scenarios,” noting its significant error reduction in challenging tasks. Reviewer dV6w appreciates the “solid theoretical foundation for GDB” and the strong empirical performance across multiple datasets. Despite concerns from Reviewer znC7 about the experimental setup and missing baselines, the authors provided extensive clarifications, including comparisons to additional relevant works and ablation studies to demonstrate the framework’s efficacy. Reviewer U4NU acknowledges the novelty of applying diffusion bridges to bridge geometric states, albeit suggesting that related works could be better discussed to strengthen the contributions. On balance, the paper presents a significant advancement in the field with robust theoretical backing and promising empirical results, meriting acceptance.